# GIMLET: A Unified Graph-Text Model for Instruction-Based Molecule Zero-Shot Learning

**Haiteng Zhao**[1], **Shengchao Liu**[2], **Chang Ma**[3], **Hannan Xu**[4],
**Jie Fu**[5], **Zhi-Hong Deng**[1], **Lingpeng Kong**[3], **Qi Liu**[3]
[1] Peking University [2] Mila [3] The University of Hong Kong
[4] University of Oxford [5] Hong Kong University of Science and Technology

## Abstract

Molecule property prediction has gained significant attention in recent years. The main bottleneck is the label insufficiency caused by expensive lab experiments. In order to alleviate this issue and to better leverage textual knowledge for tasks, this study investigates the feasibility of employing natural language instructions to accomplish molecule-related tasks in a zero-shot setting. We discover that existing molecule-text models perform poorly in this setting due to inadequate treatment of instructions and limited capacity for graphs. To overcome these issues, we propose GIMLET, which unifies language models for both graph and text data. By adopting generalized position embedding, our model is extended to encode both graph structures and instruction text without additional graph encoding modules. GIMLET also decouples encoding of the graph from tasks instructions in the attention mechanism, enhancing the generalization of graph features across novel tasks. We construct a dataset consisting of more than two thousand molecule tasks with corresponding instructions derived from task descriptions. We pretrain GIMLET on the molecule tasks along with instructions, enabling the model to transfer effectively to a broad range of tasks. Experimental results demonstrate that GIMLET significantly outperforms molecule-text baselines in instruction-based zero-shot learning, even achieving closed results to supervised GNN models on tasks such as toxcast and muv. [1]

## 1 Introduction

Molecule machine learning has gained significant attention in recent years [10, 52, 85, 77], including tasks like property prediction [14, 21], molecule design [81, 35, 28, 27], and others, which have broad applications in biomedical research. The primary method of molecule tasks is graph machine learning [21], where graphs are employed to represent molecule topology. Presently, graph machine learning approaches for molecules mainly follow the pretraining and finetuning paradigm [26, 60, 40, 84]. After the pretraining on large molecule corpus, models are able to encode informative molecule representations and generalize to downstream tasks by supervised finetuning.

One limitation of the supervised finetuning approach is the requirement on labeled data to acquire task-specific features for prediction on downstream tasks. This requirement poses a challenge in scenarios where obtaining labeled data is arduous or costly, especially in the field of molecules, and the training cost also restricts the efficiency of downstream tasks. Additionally, the supervised method exclusively relies on labeled data for acquiring task information, and is therefore unable to incorporate other additional information about the task. For instance, there exists a wealth of information regarding molecule assay tasks often provided in textual descriptions. Unfortunately, the supervised method fails to leverage such information, thereby restricting its flexibility.

---

[1]The code, model, and data are available at https://github.com/zhao-ht/GIMLET.

37th Conference on Neural Information Processing Systems (NeurIPS 2023).

In this work, we propose to investigate the feasibility of employing instructions to accomplish molecule-related tasks in a zero-shot setting. The instructions, also referred to as prompts [50, 6], constitute natural language descriptions of the task to be executed. This is inspired by recent progress in natural language processing, where the large language models possess strong generalization performance to unseen tasks by following instructions [71, 47]. This approach eliminates the need for labeled data and leverages the textual knowledge available for downstream tasks.

Several studies have explored the molecule-text language models, employing either autoregressive pretraining using language models on both the text and molecule strings (SMILES) [16, 85, 64], or contrastive pretraining of molecule and text where molecules are encoded by Graph Neural Networks (GNN) [17, 59, 39, 56]. However, these works lack the investigation into instruct-based zero-shot for downstream tasks, and our experiments have empirically demonstrated their limited performance.

We conjecture that the constraints of current molecule-text language models are mainly imposed by their inadequate treatment of instructions during pretraining and their limited capacity to represent molecule structures. First, the pretraining corpora of these models lack task description about descriptions for abundant molecule tasks as well as the supervised signal. They mainly address the capacity of general molecule-related text understanding, which may be insufficient to capture details of complex instructions to molecule properties. Second, these models' capacity is limited in representing graph structures of molecules. Molecular tasks heavily rely on understanding the structure information, while existing multimodal methods either encode SMILES sequence representation of molecules by language model or encode molecular graphs into dense vector representation using GNN, which are not conducive for language models to gain a deep understanding of graph features.

To facilitate natural language instruction-based graph learning for zero-shot molecule tasks, we propose an innovative structured language model called GIMLET (**G**raph **I**nstruction based **M**olecu**L**e z**E**ro-sho**T** learning). First, GIMLET extends large language models for both graph and text data by applying transformer mechanisms with generalized position embedding, ensuring strong capacity for both the learning of graph structure representations and the executing of instruction texts without introducing additional graph encoding modules. We further enhance the model by decoupling the encoding of the graph from specific tasks via employing attention masks. This allows for improved generalization of graph features across novel tasks.

In addition to the advanced model architecture, our approach incorporates instruction-based pretraining on an extensive dataset of molecule tasks with textual instructions. We construct a dataset comprises of 2K tasks accompanied by corresponding instructions tailored for instruction-based molecule zero-shot learning framework. Throughout the pretraining process, molecule graphs are paired with natural language instructions for molecule-related tasks, and the supervision signal is provided to train the model in executing various molecule tasks. This methodology empowers GIMLET to comprehend specific instructions for downstream tasks expressed in natural language and transfer acquired knowledge to a broad range of tasks effectively.

Our experimental results show that GIMLET outperforms molecule-text baselines by a substantial margin in instruction-based zero-shot learning, which is closed to supervised GNN on tasks like muv and toxcast. Additionally, GIMLET also exhibits impressive performance in few-shot learning and demonstrates robustness to instruction. In summary, our contributions are as follows:

- We present comprehensive investigations of natural language instruction-based graph zero-shot learning for molecule tasks, to reduce reliance on labeled data and leverage task textual knowledge. To accomplish this framework, we construct a molecule dataset consisting of two thousand tasks with instructions derived from task descriptions, which is open-sourced.

- We propose GIMLET, which extends language models to handle graph and text data. By applying the transformer mechanism with generalized position embedding and decoupled attention, our model learns graph structure representations and executes instruction texts without additional graph encoding modules. Instruction-based pretraining is applied for GIMLET to comprehend specific instructions expressed in natural language and transfer to a broad range of zero-shot tasks.

- Our experiment results outperform other molecule-text models by a substantial margin and demonstrate promising results in instruction-based zero-shot molecule tasks, demonstrating the viability of this framework. Additionally, GIMLET exhibits impressive performance in few-shot learning and demonstrates robustness to instruction.

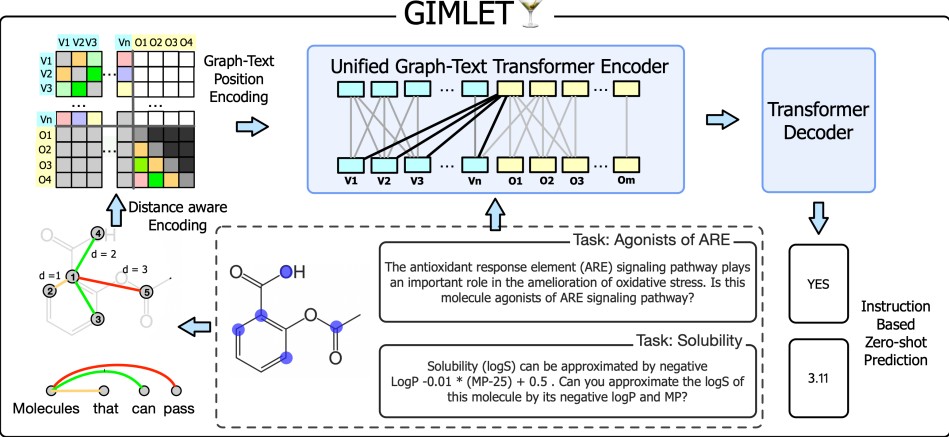

Figure 1: Our framework handles molecule tasks in the zero-shot fashion by natural language instruction. Within GIMLET, we employ distance-based joint position embedding to encode graphs and instruction texts. Additionally, we utilize attention masks to decouple the graph encoding process.

## 2 Related Work

**Molecule Representation Learning** One approach for molecular representation learning is utilizing language models to acquire molecular representations based on Simplified Molecular Input Line Entry System (SMILES) strings [68, 10]. Toward stronger capability to incorporate pertinent substructure information, Graph Neural Networks (GNNs) are proposed to model molecules as graphs [21, 83, 26, 36]. Existing GNNs follow the message-passing paradigm and suffer from problems like long-range dependency vanishing and over-smoothing. Recently, graph transformer [52, 80] has been proposed to better encode structures of graphs, illustrating effectiveness in molecule tasks [42, 31, 8, 88].

**Molecule Pretraining** To fully explore the inherent structural information of molecules on a large scale, significant efforts have been made to molecular pretraining. Supervised pretraining is commonly used for learning useful representations [26, 80, 62]. As for unsupervised pretraining, one approach involved the generative strategy on molecular SMILES strings [68, 25, 10, 3, 53] and graph [26, 37, 38, 52, 87], which was followed by recent works adopting the contrastive paradigm to distinguish augmented views of the same graph and other graphs [67, 60, 24, 83, 82, 63, 78, 19, 61, 70, 76, 69, 40].

Besides the structure-only pretraining, a few recent works incorporate natural language into molecule pretraining. One class of method is the SMILES-based language model, including KVPLM [85] and MolT5 [16], which use SMILES strings and text for joint representation and translation. Another work Galactica [64] explored the multi-task molecule task learning with instruction. Some other works acquire advanced representations for molecules by GNN, such as Text2Mol [17], MoMu [59], MoleculeSTM [39], and CLAMP [56], trained by contrastive learning between molecule graph and text description for molecule retrieval and caption tasks. MoleculeSTM and CLAMP explored molecule editing and property prediction with text. However, previous works lack the investigation into instruct-based zero-shot scenarios for complex molecule tasks like property prediction.

**Instruction-based Zero-Shot Learning** Instruction-based zero-shot learning is an innovative approach that leverages natural language instructions to enable neural models to solve a variety of tasks [50, 6, 55, 18, 89, 44, 45, 49]. To enhance the model's ability to follow instructions, some researchers have employed instruction-based pretraining techniques [54, 71, 12, 47], which explicitly train language models to solve tasks with instructions. Besides natural language processing, instruction-based zero-shot learning is also studied in multimodal domains like images [4, 9, 1, 32].

## 3 Method

### 3.1 Problem Formulation and Framework Overview

The molecule task is denoted as $\tau$ and consists of a set of graph data and their corresponding labels $(G_i, y_i^\tau)$. A molecule graph $G = (N, E, v, e)$ includes nodes $N = \{1, \ldots, n\}$ and edges $E \subset N \times N$, while $v$ and $e$ represent node features and edge features, respectively. The label $y^\tau$ can take various

forms, such as class labels or numerical values, depending on the type of task. The instruction of $\tau$ is denoted as $T^\tau$, which is a sentence $[o_1, \ldots, o_m]$ describing the task.

In the supervised approach, a model $F_\theta$ predicts the label $\hat{y}$ given a graph $G$, i.e., $\hat{y} = F_{\theta^\tau}(G)$, by supervised finetuning individually for each downstream task $\tau$. The limitation is that it relies on labeled data to learn the corresponding parameters and output modules, and finetuning does not enable the model to effectively utilize extra task-related knowledge.

To overcome these limitations, our framework leverages natural language instructions $T^\tau$ to provide the model with task information, as illustrated in Figure 1. Our zero-shot graph learning framework incorporates molecule graphs $G$ and task instructions $T^\tau$ into a graph-text language model GIMLET and decodes the output as text uniformly for different tasks, i.e. $\hat{y}_{\text{str}} = \text{GIMLET}(G, T^\tau)$, where $\hat{y}_{\text{str}}$ is the label string. This description-based instruction framework empowers the model to handle a wide range of molecule tasks as long as they can be described in natural language, and the uniform text decoding approach accommodates various types of tasks including classification or regression.

Previous pretrained molecule-text models perform poorly in our framework due to the inadequate treatment of instructions and limited capacity for graphs. Our method addresses these from two aspects: First, we propose a unified language model GIMLET for both graph and text, towards the stronger capacity to represent molecules and instructions; Second, we adopt instruction-based pretraining to GIMLET, enabling generalization to new tasks based only on the instructions.

### 3.2 Unified Graph-Text Transformer

The common method for multimodal language models is obtaining the feature of the other modality by applying an additional encoding module, then integrating it into the language model, as in other molecule language models with GNN encoder [17, 59, 56]. The individual encoding module benefits for decoupling the graph feature encoding from instructions, i.e., the features of graph data can be independent of task instructions in the early encoding stage, helping for generalization when the distribution of tasks changes.

However, for the molecule-text model, individual pre-encoding modules present problems. First, graph learning relies on structure information, but the dense vectors encoded by GNN have a limited capacity to carry structure information. Furthermore, training the additional module is difficult due to the increased layers, since deep transformers have vanishing gradients in early layers [34, 2]. Lastly, the additional modules increase parameters and training costs.

To overcome these issues, we propose a novel approach GIMLET which not only directly unifies the standard language model for graph and text *without* introducing additional graph encoder module, but also remains the decoupled graph encoding for better generalization.

Given graph $G$ and text input $T$, to utilize GIMLET, we represent the graph nodes and text tokens as tokens. The resulting hidden states are denoted as $H = [h_1, \ldots, h_n, h_{n+1}, \ldots, h_{n+m}]^T \in \mathbb{R}^{(n+m) \times d_h}$ for corresponding $n$ graph nodes and $m$ text tokens.

In this study, we choose T5 [50] as the backbone language model, due to the encoder-decoder architecture suitable for non-sequential encoding and text output. It utilizes the relative position embedding method [57] to represent sequential structure. In attention layer Attn with parameter $W^V, W^Q, W^K \in \mathbb{R}^{d_h \times d_k}$ and $W^O \in \mathbb{R}^{d_k \times d_h}$, relative position embedding for i-th and j-th token is formalized as

$$\hat{A}_{ij} = \frac{(h_i W^Q)(h_j W^K)^T}{\sqrt{d_k}} + b(i, j), A = \text{softmax}(\hat{A}), \quad \text{Attn}(H) = AHW^V W^O, \tag{1}$$

where $b(i, j)$ is embedding of the relative distance between $i$ and $j$, i.e. $i - j$ for sequence. For graph-text joint data, we construct the position embedding $b(i, j)$ in E.q. 1 by the conjunction of different types of distances. For the relative position of graph nodes, we adopt the graph shortest distance, which has been widely used in the literature of graph transformer [80, 11, 48]. We also use unidirectional constraint in attention to decouple the graph encoding from the instruction. The overall form of position embedding for GIMLET is:

$$b(i, j) = b^D_{\text{POS}(i,j)} + b^M_{i,j} + \underset{k \in \text{SP}(i,j)}{\text{Mean}} b^E_{e_k}, \tag{2}$$

where $b^D$, $b^M$, and $b^E$ are position bias, masking, and path embedding, individually. The relative position POS in $b^D_{\text{POS}(i,j)}$ is defined as the conjunction of different types of distances between tokens,

which allows for the effective encoding of both graph and text data, as well as their interaction:

$$\text{POS}\,(i,j) = \begin{cases} i - j & \text{if } n + 1 \leq i, j \leq n + m \\ \text{GRAPH SHORTEST DISTANCE}(i,j) & \text{if } 1 \leq i, j \leq n \\ <\text{CROSS}> & \text{otherwise} \end{cases}, \tag{3}$$

where <CROSS> is a special distance token held out for cross distance between graph and text tokens. $b_{i,j}^M$ aims to represent the cross mask used to decompose graph encoding from text instructions. It imposes a unidirectional constraint from the graph to the text:

$$b_{i,j}^M = -\infty \text{ if } i \leq n \text{ and } j > n \quad \text{otherwise } 0 \tag{4}$$

With the unidirectional constraint, graph tokens are limited to attending only to other graph tokens. On the other hand, instruction tokens have the ability to receive information from both instructions and graphs. This approach allows us to separate the encoding of graphs from instructions, enabling instructions to selectively utilize graph features for various downstream tasks.

Finally, $\text{Mean}_{k \in \text{SP}(i,j)}\, b_{e_k}^E$ is the mean pooling of the edge features $b_{e_k}^E$ in the shortest path $\text{SP}(i,j)$ between node $i$ and $j$, which is only defined between graph node tokens, as used along with graph shortest distance in [80].

The generalized position embedding is applied to the encoder transformer. During decoding, the decoder generates the answer based on the text features outputted by the encoder.

GIMLET unifies graph and text data by a single language model, which has the following merits: **(i)** In comparison to additional encoding module methods, GIMLET not only avoids the challenges of training additional front layers but also provides a stronger capacity for handling graph data by introducing graph inductive bias to the whole transformer. **(ii)** Our method leverages both the existing knowledge within the language model and facilitates learning to execute instruction-based graph tasks through standard instruction-based learning on the language model. This approach eliminates the need for additional modeling costs. **(iii)** The decomposed encoding of graph data retains the advantages of the individual encoding module, reducing task disturbance and ensuring that data features remain independent of task instructions. This enhances generalization for novel instructions. We validate these claims in experiments Subsection 4.3.

## 3.3 Pretraining and Datasets

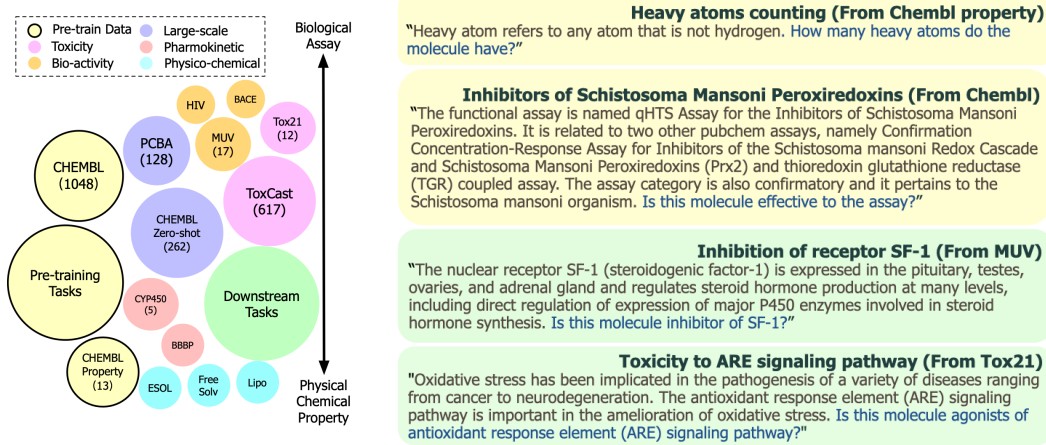

Figure 2: (Left) Illustration of datasets. Circle size corresponds to task number. Tasks are organized by category. Tasks on the top are more related to biological assay, on the bottom need more chemical and physical properties. GIMLET is trained on pretraining tasks, then tested on downstream tasks in the zero-shot setting. (Right) Our task instructions contain task explanations and questions.

**Pretraining** Our approach leverages the generalization capabilities acquired through learning from the instructions provided during pretraining, where comprehensive linguistic information and knowledge related to molecular tasks are learned from the provided instructions. The pretraining process is

conducted in a supervised manner using task labels. The loss function for supervised pretraining can be formalized as follows:

$$L = \frac{1}{\sum_\tau |\tau|} \sum_\tau \sum_{(G_i, y_{str,i}^\tau) \in \tau} -\frac{1}{|y_{str,i}^\tau|} \log P_{\text{GIMLET}}(y_{str,i}^\tau | G_i, T^\tau),$$ (5)

where $P_{\text{GIMLET}}$ is the model likelihood for the label string, $|y_{str,i}^\tau|$ represents the label string token number aiming to normalize the loss of long sequences, and $|\tau|$ is the task sample number. The details of pretraining and downstream zero-shot testing are in Appendix.

**Pretraining Dataset** To effectively pretrain GIMLET for downstream tasks, it is crucial to include a large number of tasks to provide the model with an extensive task corpus. To this end, we select Chembl [20] as the pretraining dataset, which is widely used for supervised graph pretraining [26, 62]. It consists of 1,310 prediction target labels from biological assays for drug discovery. We divided 80% of the tasks and 80% of the molecules in random for pretraining, while the remaining non-overlapping tasks were reserved for zero-shot testing. Additionally, we constructed the Chembl-property dataset, which encompasses various physical and chemical properties available in the Chembl database [20] like molecule weights, log p values, hydrogen bound acceptor numbers, etc. Full details are in Appendix. We validate the effect of Chembl biological tasks and Chembl-property physico-chemical tasks in pretraining in Subsection 4.3.

**Downstream Dataset** We target a large range of molecule tasks, as shown in Figure 2. First, we include large-scale datasets PCBA [72], which contains 128 biological assay tasks with abundant molecules. We also include the hold-out zero-shot set of Chembl, noted as Chembl Zero-Shot. These two datasets form a large-scale dataset. We also target tasks from MoleculeNet [75], a popular benchmark for molecule properties prediction. We adopt the dataset categorization method in [58] which classifies MoleculeNet datasets into four categories: Physico-chemical tasks, Bio-activity tasks, Toxicity tasks, and Pharmacokinetic tasks. Additional dataset CYP450 [33] is also included in the Pharmacokinetic tasks. These tasks cover diverse aspects of molecule properties, posing a significant challenge for a unified model to simultaneously handle them in a zero-shot fashion.

**Instructions** To provide essential background information and context for each task, we include task explanations and descriptions in our instructions. The description covers a wide range of aspects, including the family, function, and mechanism of the assay target, the assay experiment setting, the approximation method used for determining the property, and others. See examples in Figure 2. Instructions for all tasks are available in the code file. The task explanation is primarily sourced from websites and databases that introduce and compile the respective datasets, or relevant papers. Details are in Appendix. The descriptions are then concatenated with relevant questions as instructions. These instructions are subsequently reviewed and validated by a professional biology Ph.D. student.

## 4 Experiment

In the experiments, we investigate the following inquiries: (**i**) Can GIMLET effectively handle zero-shot molecule property tasks by instructions? (**ii**) Can GIMLET performs better by few-shot learning? (**iii**) What impact does model architecture have on the performance of GIMLET? (**iv**) How does pretraining affect the performance of GIMLET? (**v**) How does the form of instruction influence GIMLET for molecule zero-shot learning?

### 4.1 Instruction-Based Zero-Shot Learning

**Baselines** In the zero-shot setting, we compare GIMLET with three molecule-text models: SMILES-base language model KVPLM [85], Galactica [64], and GNN-language model MoMu [59]. Other molecule-text models [16, 17, 39] either haven't released parameters or are difficult to handle zero-shot setting. Notably, the pretraining of Galactica includes the MoleculeNet datasets, which is thus not strictly zero-shot on some of our tasks. We report the result of all the baselines with our zero-shot learning framework and instructions. The details of baseline evaluation are in Appendix.

To establish upper bounds for task performance, we also illustrate the supervised results of popular graph models. For GNNs, we includes GCN [30], GAT [66], and GIN [79]. We also include the graph transformer Graphormer [80]. To mitigate the impact of our pretraining, we additionally perform supervised pretraining on Graphormer using our pretraining datasets, referred to as Graphormer-p.

**Settings** Following the standard supervised setting in previous studies [26], we adopt the Scaffold split [51] with a ratio of 0.8, 0.1, 0.1 for all the datasets, and report results on the testing sets, ensuring the comparability of our results to previous works. For classification tasks, we employ ROC-AUC as the evaluation metric, while for regression tasks, we utilize RMSE.

Table 1: Zero-shot performance (ROC-AUC) over Bio-activity, Toxicity, and Pharmacokinetic tasks.

| Method | #Param | Type | bace | hiv | muv | Avg. bio | tox21 | toxcast | Avg. tox | bbbp | cyp450 | Avg. pha |
|---|---|---|---|---|---|---|---|---|---|---|---|---|
| KVPLM | 110M | | 0.5126 | 0.6120 | 0.6172 | 0.5806 | 0.4917 | 0.5096 | 0.5007 | 0.6020 | 0.5922 | 0.5971 |
| MoMu | 113M | | 0.6656 | 0.5026 | 0.6051 | 0.5911 | 0.5757 | 0.5238 | 0.5498 | 0.4981 | 0.5798 | 0.5390 |
| Galactica-125M | 125M | Zero Shot | 0.4451 | 0.3671 | 0.4986 | 0.4369 | 0.4964 | 0.5106 | 0.5035 | **0.6052** | 0.5369 | 0.5711 |
| Galactica-1.3B | 1.3B | | 0.5648 | 0.3385 | 0.5715 | 0.4916 | 0.4946 | 0.5123 | 0.5035 | 0.5394 | 0.4686 | 0.5040 |
| GIMLET (Ours) | 64M | | **0.6957** | **0.6624** | **0.6439** | **0.6673** | **0.6119** | **0.5904** | **0.6011** | 0.5939 | **0.7125** | **0.6532** |
| GCN | 0.5M | | *0.736* | *0.757* | *0.732* | *0.742* | *0.749* | *0.633* | *0.691* | *0.649* | 0.8041 | 0.7266 |
| GAT | 1.0M | | *0.697* | *0.729* | *0.666* | *0.697* | *0.754* | *0.646* | *0.700* | *0.662* | 0.8281 | 0.7451 |
| GIN | 1.8M | Supervised | *0.701* | *0.753* | *0.718* | *0.724* | *0.740* | *0.634* | *0.687* | *0.658* | 0.8205 | 0.7392 |
| Graphormer | 48M | | 0.7760 | 0.7452 | 0.7061 | 0.7424 | 0.7589 | 0.6470 | 0.7029 | 0.7015 | 0.8436 | 0.7725 |
| Graphormer-p | 48M | | 0.8575 | 0.7788 | 0.7480 | 0.7948 | 0.7729 | 0.6649 | 0.7189 | 0.7163 | 0.8877 | 0.8020 |

Table 2: Zero-shot performance (ROC-AUC) over large scale molecule tasks.

| Method | Chembl Zero-Shot | PCBA |
|---|---|---|
| KVPLM | 0.4155 | 0.4811 |
| MoMu | 0.5002 | 0.5150 |
| Galactica-125M | 0.6461 | 0.4800 |
| Galactica-1.3B | 0.4818 | 0.5202 |
| GIMLET (Ours) | **0.7860** | **0.6211** |

Table 3: Zero-Shot performance (RMSE) on Physical-chemical datasets.

| Method | Type | ESOL | Lipophilicity | FreeSolv | Avg. phy |
|---|---|---|---|---|---|
| KVPLM | | - | - | - | - |
| MoMu | Zero Shot | - | - | - | - |
| GIMLET (Ours) | | 1.132 | 1.345 | 5.103 | 2.527 |
| GCN | | 1.331 | 0.760 | 2.119 | 1.403 |
| GAT | | 1.253 | 0.770 | 2.493 | 1.505 |
| GIN | Supervised | 1.243 | 0.781 | 2.871 | 1.632 |
| Graphormer | | 0.901 | 0.740 | 2.210 | 1.284 |
| Graphormer-p | | 0.804 | 0.675 | 1.850 | 1.110 |

**Results** We report the result of different types of downstream tasks in Table 1. The result of GIN, GCN and GAT for MoleculeNet are from [26] which we mark by italic. We observe that in the zero-shot setting, GIMLET outperforms most baselines on the majority of datasets, except for bbbp where GIMLET also performs comparably to the baselines. In terms of the average performance across task categories, GIMLET outperforms all baselines, demonstrating the effectiveness of our method for instruction-based zero-shot molecule tasks. It is worth noting that some of the baselines also achieve results on certain tasks. For example, KVPLM works on hiv, muv and bbbp, and MoMu works on tox21 and bace, showing our instruction-based molecule zero-shot learning method is a general framework to probe knowledge in molecule-text models.

Comparing our zero-shot performance to the supervised results, we observe that GIMLET achieves performance close to those of GNNs on several datasets, like bace, muv, toxcast, and bbbp. This demonstrates that GIMLET is able to solve molecule tasks in the zero-shot fashion nicely.

The results for large-scale molecule tasks are presented in Table 2. As depicted, the baselines struggle to handle these tasks. Our GIMLET not only successfully transfers to the Chembl Zero-Shot splits, where both the tasks and graphs were unseen during pretraining, but also demonstrates strong generalization performance on the PCBA benchmark.

The results of regression tasks are shown in Table 3. The scatter plots of the regression are in Appendix. It is worth noting that regression tasks pose greater challenges in the zero-shot setting than classification tasks, because it is difficult to determine unseen physico-chemical properties using only natural language, and the output space is also vast. Notably, the zero-shot baselines fail to perform the regression tasks due to their inability to output correctly formatted numbers. In contrast, GIMLET generates correctly formatted numbers for over 98% regression testing samples in all the tasks and showcases the potential of zero-shot regression tasks.

### 4.2 Instructions-Based Few-Shot Finetuning

We apply few-shot instruction-based tuning on the downstream tasks, to examine whether GIMLET exhibits improved performance in the presence of low-resource data. Notably, for datasets with more than one task, we do few-shot learning for every single task individually. We first split datasets into training, validation, and testing sets in the same as the zero-shot setting. Then $K$ samples for each class are randomly sampled from the training set as the few-shot examples, where $K$ is the few-shot number. We report the result of the best validation model on the testing set. The input is the

same as the zero-shot setting, including molecule data and instructions. We only tune the last linear layer, to inspect whether the feature is discriminative and linear separable. Linear tuning is also low resource costing and avoids overfitting. The last linear mapping to vocabulary is tuned for GIMLET and KVPLM. For MoMu, we tune the last linear layer of the projection head for features.

The result is shown in Figure 3, and plots for each dataset are in Appendix. GIMLET outperforms other baselines consistently, exhibiting improved performance with an increasing number of few-shot samples. The performance of GIMLET also approaches the supervised GIN with only limited samples. Baselines also achieve some performance gain in the few-shot training but remain beaten by GIMLET, and the improvements are not stable, fluctuating with the few-shot number. The few-shot results demonstrate the high-quality representation learned by GIMLET, which is well-suited for specific tasks, highlighting the potential of GIMLET in scenarios beyond zero-shot learning.

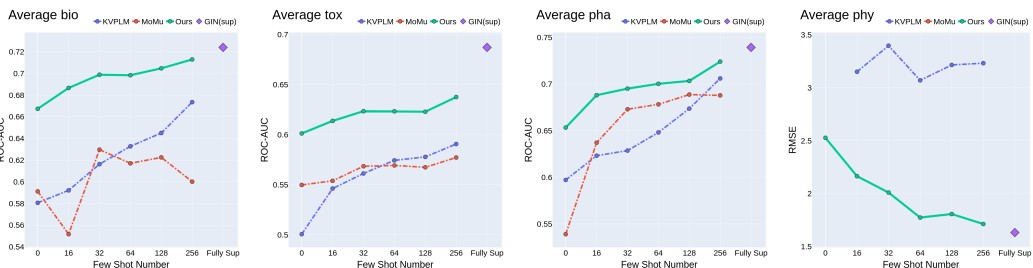

Figure 3: Few shot performance. Higher is better for bio, tox, and pha, and lower is better for phy.

## 4.3 Ablation and Exploration Studies

**Effect of Model Design** We investigate components of GIMLET to highlight their benefits. Specifically, we focus on two aspects: **(a)** To measure the effectiveness of the unified transformer method, we compare it to the variant version which obtains graph embedding by an individual GIN as a token in text, a common method in multimodal transformers. **(b)** We ablate graph decoupling encoding by complete global attention. We conduct pretraining and downstream zero-shot testing in the same setting as our method.

Table 4: Ablation study on GIMLET module.

| Method | bace | hiv | muv | Avg. bio | tox21 | toxcast | Avg. tox | bbbp | cyp450 | Avg. pha |
|---|---|---|---|---|---|---|---|---|---|---|
| w.o. unifying | 0.4319 | 0.6133 | 0.6067 | 0.5506 | 0.5922 | 0.5537 | 0.5730 | 0.5309 | 0.6206 | 0.5758 |
| w.o. decoupling | 0.6458 | 0.6406 | 0.5421 | 0.6095 | **0.6306** | **0.5954** | **0.6130** | 0.5666 | 0.6320 | 0.5993 |
| GIMLET | **0.6957** | **0.6624** | **0.6439** | **0.6673** | 0.6119 | 0.5904 | 0.6011 | **0.5939** | **0.7125** | **0.6532** |

The comparison is shown in Table 4. Compared to GIMLET, w.o. unifying perform worse on all the datasets, especially on Bio-activity and Pharmacokinetic Tasks. Next, the w.o. decoupling variation performs worse than GIMLET on most datasets. The decoupling significantly improves performance on Bio-activity and Pharmacokinetic tasks and is also comparable on Toxicity tasks. This proves our claims that the unified graph-text transformer not only avoids additional modules but also has a strong capacity for encoding graph data and generalizing across tasks.

**Effect of Pretraining** Our pretraining includes two large types of pretraining tasks, including Chembl bioactivity tasks and Chembl Property physico-chemical tasks. To validate the influence of each task type, we performed ablation experiments where each of them was excluded separately. The results are shown in Table 5, and the detailed result is in Appendix. Unsurprisingly, Chembl is essential for downstream molecule assay tasks, and Chembl property plays a crucial role in Physico-chemical tasks. However, the results also reveal that Chembl positively affects downstream Physico-chemical tasks, and Chembl property benefits Bio-activity tasks, Toxicity tasks, and Pharmacokinetic tasks. The results demonstrate the positive transfer of pretraining tasks on a diverse range of downstream tasks, spanning various types and domains.

**Robustness to Instruction** We explore whether GIMLET is robust to the instructions. We rephrase our instructions by GPT-3.5-turbo for testing. Each instruction is rephrased using four types of requests:

Table 5: Ablation study on GIMLET pretraining.

| Method | Avg. bio ↑ | Avg. pha ↑ | Avg. tox ↑ | Avg. phy ↓ |
|---|---|---|---|---|
| bioactivity assay only | 0.6402 | 0.6071 | 0.5676 | - |
| physico-chemical only | 0.4894 | 0.5454 | 0.4748 | 2.6178 |
| both | **0.6673** | **0.6532** | **0.6011** | **2.5266** |

rewriting, detailing, expanding, and shortening. The prompts and examples are provided in Appendix. We plot the performance for each type of augmentation, as well as the average standard variation for each task in Figure 4. As shown, GIMLET is more robust than baselines on most tasks. This shows that our instruction-based pretaining enables GIMLET to focus on the task rather than the specific language form.

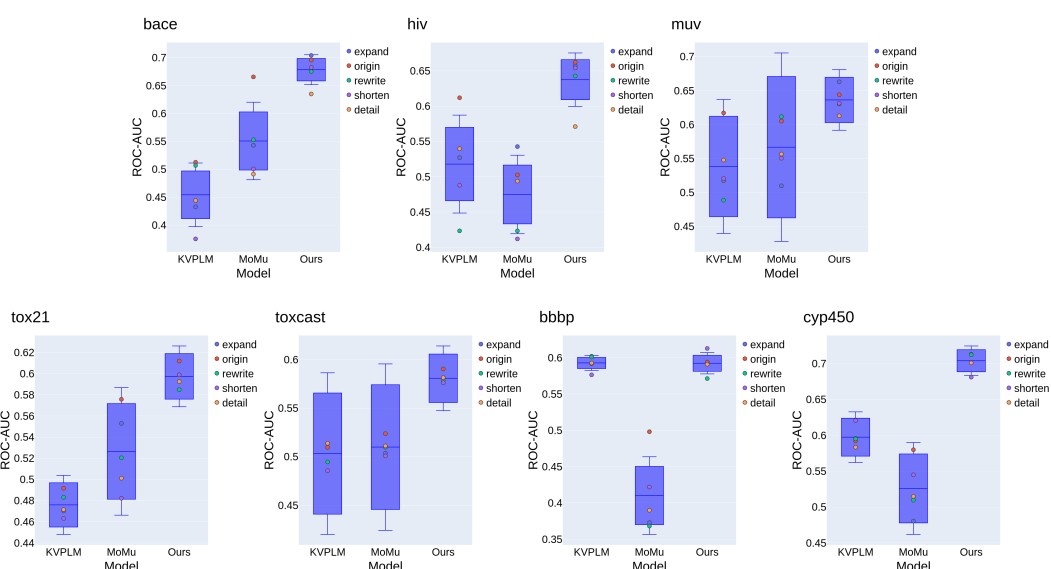

Figure 4: Robustness to instruction.

**Instruction Interpretability** We ablate the explanation of the instructions in downstream tasks, to validate whether the explanation in instructions helps GIMLET perform downstream tasks. Without explanation, only the task name and question are provided to the model. The examples of ablated instructions are available in Appendix. The results presented in Table 6 demonstrate a significant drop in performance when task explanations are not included. This finding supports the effectiveness of the explanation and highlights the model's ability to comprehend the explanation of tasks.

Table 6: Ablation study on GIMLET instructions.

| Method | bace | hiv | muv | Avg. bio | tox21 | toxcast | Avg. tox | bbbp | cyp450 | Avg. pha |
|---|---|---|---|---|---|---|---|---|---|---|
| name only | 0.5416 | 0.6132 | **0.6441** | 0.5996 | 0.5809 | 0.5279 | 0.5544 | 0.4871 | 0.6669 | 0.5770 |
| + explanation | **0.6957** | **0.6624** | 0.6439 | **0.6673** | **0.6119** | **0.5904** | **0.6011** | **0.5939** | **0.7125** | **0.6532** |

## 5   Conclusion and Discussion

In this work, we propose nature language instruction-based graph zero-shot learning for molecule tasks, and construct a molecule dataset consisting of two thousand tasks with instructions derived from task descriptions. We propose GIMLET, which extends large language models to handle graph and text data by applying the transformer mechanism with generalized position embedding and decoupled attention. Instruction-based pretraining is applied for GIMLET. Experiments demonstrate promising results in zero-shot learning, exhibit strong robustness to the instruction, and can be further improved by few-shot tuning. We do not consider the tasks with structured output like molecule generation in this study, which is left for further work.

# 6 Ethical Consideration

The work presented here is centered on a paradigm shift in molecule property prediction tasks, transitioning from traditional supervised learning to instruction-based zero-shot learning. Due to the fact that molecule property prediction has been widely explored in prior research, the direct societal impacts may appear limited. However, indirect negative impacts could be caused by excessive reliance on algorithms. We contend that a combination of model predictions and experimental validation is essential for practical implementation.

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

# A Framework

## A.1 Details of Datasets and Instructions

Table 7: Data Overview

| Splitting | Data Class | Dataset | No. of Molecules | No. of Tasks | Task Metric | Task Type |
|---|---|---|---|---|---|---|
| Pretraining | Bioactivity assay | ChEMBL bioassay activity dataset | 365065 | 1048 | ROC_AUC | Classification |
| | Physico-chemical | CHEMBL Property | 365065 | 13 | RMSE | Regression |
| Downstream Zero-Shot | Large Scale | PCBA PubChem HTS bioAssay | 437929 | 128 | ROC-AUC | Classification |
| | | ChEMBL Zero-Shot | 91266 | 262 | ROC_AUC | Classification |
| | Pharmacokinetic | CYP inhibition | 16896 | 5 | ROC_AUC | Classification |
| | | BBBP Blood-brain barrier penetration | 2039 | 1 | ROC_AUC | Classification |
| | Bio-activity | MUV PubChem bioAssay | 93087 | 17 | ROC_AUC | Classification |
| | | BACE-1 benchmark set | 1513 | 1 | ROC_AUC | Classification |
| | | HIV replication inhibition | 41127 | 1 | ROC_AUC | Classification |
| | Toxicity | Tox21Toxicology in the 21st century | 7831 | 12 | ROC_AUC | Classification |
| | | Toxcast | 8598 | 617 | ROC_AUC | Classification |
| | Physico-chemical | ESOL Water solubility | 1128 | 1 | RMSE | Regression |
| | | FreeSolv Solvation free energy | 642 | 1 | RMSE | Regression |
| | | Lipo Lipophilicity | 4200 | 1 | RMSE | Regression |

The datasets used in our study are presented in Table 7. These datasets consist of different types of tasks related to molecule property prediction. It should be noted that during the pretraining phase, the loss function is not specific to the task types, but rather encompasses the generative loss of the language model.

We have chosen not to include certain datasets, namely SIDER and ClinTox, in our collection of datasets. The decision was based on the fact that the tasks associated with these datasets are not clearly defined and involve complex systemic phenomena, making it challenging to describe them through instructional texts. For instance, the ClinTox dataset involves determining whether drugs have passed the FDA approval, which is not an objective problem but rather a dynamic and intricate social phenomenon. The SIDER dataset focuses on describing the side effects of drugs on system organ classes, which have intricate mechanisms and a wide range of possible causes, making them difficult to be effectively conveyed through instructions.

For the Chembl property dataset that we have constructed, detailed information can be found in Table 8. These properties are sourced from the Chembl database [20] through the web API.

Table 8: Chembl property tasks and labels

| Property | Label type |
|---|---|
| Aromatic rings number | Integer |
| cx_logd distribution coefficient | Real |
| cx_logp partition coefficient | Real |
| cx_most_apka $-\log_{10}$ dissociation constant | Real |
| Molecular masses | Real |
| Hydrogen bond donor number | Integer |
| Heavy atom number | Integer |
| Lipinski's rule of five violation number | Integer |
| Polar surface area (PSA) | Real |
| Quantitative Estimate of Druglikeness (QED) | Real |
| Rule of three passes | Bool |
| Rotatable bond number | Integer |

The task explanation is primarily sourced from relevant papers, websites, or databases that introduce and compile the respective datasets. The specific sources utilized depend on the particular datasets under consideration. For Chembl tasks, we obtain task descriptions from the Chembl website. Descriptions for MoleculeNet tasks and PCBA are primarily sourced from the PubChem website. Certain datasets, such as Toxcast, include task descriptions within the dataset files. In the case of other tasks, like Chembl property and Physical-Chemical tasks, instructions are derived from Wiki or other papers. We list the instruction source in Table 9.

The description covers a wide range of aspects, including the family, function, and mechanism of the assay target, the assay experiment setting, the approximation method used for determining the property, and others. We describe regression tasks by introducing the relasionship between the

Table 9: Data sources and classes for different stages of the model

| Dataset | Instruction Source |
|---|---|
| ChEMBL Zero-Shot bioassay activity dataset | Chembl Database |
| CHEMBL Property | Wiki |
| PCBA PubChem HTS bioAssay | Pubchem Database |
| ChEMBL Zero-Shot bioassay activity dataset | Chembl Database |
| CYP PubChem BioAssay CYP 1A2, 2C9, 2C19, 2D6, 3A4 inhibition | Pubchem Database |
| BBBP Blood-brain barrier penetration | Paper [22] |
| MUV PubChem bioAssay | Pubchem Database |
| BACE-1 benchmark set | Pubchem Database |
| HIV replication inhibition | Paper [46] |
| Tox21 Toxicology in the 21st century | Pubchem Database |
| Toxcast | Toxcast file |
| ESOL Water solubility | Paper [73] |
| FreeSolv Solvation free energy | Paper [7] |
| Lipo Lipophilicity | Wiki |

task property and other properties, i.e. how to estimate these properties by other ones. However, this method is still challenging due to the model's capacity to understand complex mathematical relationships.

The instructions for each task are generated automatically by conducting searches on the databases and summarizing the descriptions. We use a mixture strategy of summarizing, combining template-based summarizing and GPT-3.5-turbo-based summarizing methods. The GPT-3.5-turbo-based summarizing method is applied by the prompt 'Summarize the assay: \n {*Descriptions to be summarized*}'.

The resulting instructions are then concatenated with relevant questions. These instructions are subsequently reviewed and validated by a professional biology Ph.D. student and slightly modified if necessary.

We then list the instructions of each dataset. For datasets with more than one task, we only list the instruction of one task as an illustration.

Chembl

```
"The assay is PUBCHEM_BIOASSAY: qHTS Assay for Activators of
Human Muscle isoform 2 Pyruvate Kinase. (Class of assay:
confirmatory)  , and it is Direct single protein target
assigned . The assay has properties: assay category is
confirmatory ; assay organism is Homo sapiens ; assay type
description is Functional . Is the molecule effective to this
assay?"
```

Chembl property

```
"The partition coefficient, abbreviated P, is defined as a
particular ratio of the concentrations of a solute between the
two solvents (a biphase of liquid phases), specifically for
un-ionized solutes, and the logarithm of the ratio is thus Log
P. When one of the solvents is water and the other is a
non-polar solvent, then the log P value is a measure of
lipophilicity or hydrophobicity. The defined precedent is for
the lipophilic and hydrophilic phase types to always be in the
numerator and denominator respectively. What is the logarithm
of the partition coefficient of this molecule?"
```

PCBA

```
"The assay tests the inhibition of ALDH1A1 activity using
propionaldehyde as an electron donor and NAD+ as an electron
acceptor. The conversion of NAD+ to NADH is measured via an
```

increase in fluorescence intensity to determine enzyme
activity. ALDH1A1 plays critical roles in the metabolic
activation of retinoic acid and may be a target for inhibitor
development in metabolic diseases. Is the molecule effective
to this assay?"

CYP450

"Find molecules that can effectively inhibit Cytochrome P450
(CYP450) enzymes, particularly CYP1A2, to help reduce the risk
of adverse drug events and drug-drug interactions caused by
CYP450-mediated metabolic pathways. Consider the various
CYP450 inhibition mechanisms such as occupying active sites or
weakening enzyme activity, while keeping in mind the potential
for increased side effects due to elevated blood drug
concentrations. Is this molecule effective to this assay?"

BBBP

"In general, molecules that passively diffuse across the brain
blood barrier have the molecular weight less than 500, with a
LogP of 2-4, and no more than five hydrogen bond donors or
acceptors. Does the molecule adhere to the three rules or not?"

MUV

"Protein kinase A (PKA) is an ubiquitous serine/threonine
protein kinase and belongs to the AGC kinase family. It has
several functions in the cell, including regulation of immune
response, transcription, cell cycle and apoptosis. PKA is a
cAMP dependent enzyme that exists in its native inactive form
as a 4 subunit enzyme with two regulatory and two catalytic
subunits. Binding of cAMP to the regulatory subunit leads to
the disassembly of the complex and release of now active
catalytic subunits. Is this molecule inhibitor of PKA?"

BACE

"BACE1 is an aspartic-acid protease important in the
pathogenesis of Alzheimer's disease, and in the formation of
myelin sheaths. BACE1 is a member of family of aspartic
proteases. Same as other aspartic proteases, BACE1 is a
bilobal enzyme, each lobe contributing a catalytic Asp
residue, with an extended active site cleft localized between
the two lobes of the molecule. The assay tests whether the
molecule can bind to the BACE1 protein. Is this molecule
effective to the assay?"

HIV

"Human immunodeficiency viruses (HIV) are a type of
retrovirus, which induces acquired immune deficiency syndrome
(AIDs). Now there are six main classes of antiretroviral
drugs for treating AIDs patients approved by FDA, which are
the nucleoside reverse transcriptase inhibitors (NRTIs), the
non-nucleoside reverse transcriptase inhibitors (NNRTIs), the
protease inhibitors, the integrase inhibitor, the fusion
inhibitor, and the chemokine receptor CCR5 antagonist. Is
this molecule effective to this assay?"

Tox21

"Estrogen receptor alpha (ER aplha) is Nuclear hormone
receptor. The steroid hormones and their receptors are
involved in the regulation of eukaryotic gene expression and
affect cellular proliferation and differentiation in target
tissues. Ligand-dependent nuclear transactivation involves
either direct homodimer binding to a palindromic estrogen
response element (ERE) sequence or association with other
DNA-binding transcription factors, such as AP-1/c-Jun, c-Fos,
ATF-2, Sp1 and Sp3, to mediate ERE-independent signaling. Is
this molecule effective to this assay?"

Toxcast

"APR_HepG2_CellCycleArrest_24hr, is one of 10 assay
component(s) measured or calculated from the APR_HepG2_24hr
assay. It is designed to make measurements of cell phenotype,
a form of morphology reporter, as detected with fluorescence
intensity signals by HCS Fluorescent Imaging technology.Data
from the assay component APR_HepG2_CellCycleArrest_24hr was
analyzed into 2 assay endpoints. \nThis assay endpoint,
APR_HepG2_CellCycleArrest_24h_dn, was analyzed in the negative
fitting direction relative to DMSO as the negative control and
baseline of activity. \nUsing a type of morphology reporter,
measures of all nuclear dna for loss-of-signal activity can be
used to understand the signaling at the pathway-level as they
relate to the gene . \nFurthermore, this assay endpoint can be
referred to as a primary readout, because this assay has
produced multiple assay endpoints where this one serves a
signaling function. \nTo generalize the intended target to
other relatable targets, this assay endpoint is annotated to
the \"cell cycle\" intended target family, where the subfamily
is \"proliferation\". Is this molecule effective to this
assay?"

ESOL

"Solubility (logS) can be approximated by negative LogP -0.01
* (MPt \u2013 25) + 0.5 . Can you approximate the logS of this
molecule by its negative logP and MPt?"

FreeSolv

"The free energy of hydration can be approximated by
\u0394G_hyd = \u0394G_solv,soln - \u0394G_solv,gas + RT ln
(10^(-pKa)). Can you tell me the free energy of hydration (by
using the negative pka) of this molecule, predicted by using
\u0394G_solv and negative pka?"

Lipo

"Lipophilicity is an important feature of drug molecules that
affects both membrane permeability and solubility, measured by
octanol/water distribution coefficient (logD at pH 7.4).
What's the octanol/water distribution coefficient (logD at pH
7.4) of this molecule?"

## A.2 Details of Framework Application

In our framework, we represent the labels of various tasks as strings. For assay tasks involving classification, the labels are converted to either "Yes" or "No" based on whether the molecule has

an effect on the assay. In regression tasks, the labels are transformed into numerical strings. Integer values remain unchanged, while decimal numbers are rounded to two decimal places.

To conduct zero-shot testing on our model, we generate output sequences and extract the answer from the results. For assay classification, we consider the first token generated as the answer and use the scores for the 'Yes' and 'No' tokens to compute the ROC-AUC score for classification. In regression tasks, we extract the number from the generated sequence by performing string matching using a regular expression template: r"-?\d+\.?\d*e??\d*?". Notably, we discovered that GIMLET consistently generates results in the correct format for all classification tasks and accurately formatted numbers for over 98% of regression testing samples, without any augmentation of restriction in the vocabulary.

## A.3 Baselines Evaluation

For the baselines, we apply our instruction-based molecule zero-shot learning to their respective settings. KVPLM employs SMILES for molecule representation and utilizes masked language modeling for molecule-text data. Galactica also represents molecules using SMILES but generates the next sentence in an autoregressive manner. MoMu employs contrastive learning between the GNN-encoded molecule and the corresponding text, allowing it to score each candidate sentence for the target molecule and retrieve the best matching one. Our application of each baseline model aligns with their intended use.

It is important to note that for the baseline models, to avoid baselines generating answers in classification not in our parsing method ('Yes' and 'No'), we limit the vocabulary during generation to only include 'Yes' and 'No' in classification tasks. This restriction is achieved by utilizing the bias term in huggingface to prevent the generation of other words. However, it is worth mentioning that our model, GIMLET, does *not* require this augmentation and is able to generate the desired outputs *without* any additional constraints.

For KVPLM, we mask the answer position in the whole sentence for the model to predict. For example, for molecule CCOc1ccccc1-n1nnnc1SCC(=O)NC(=O)NCc1ccco1 and classification tasks ARE inhibitor, input to KVPLM is:

```
"CCOc1ccccc1-n1nnnc1SCC(=O)NC(=O)NCc1ccco1
Oxidative stress has been implicated in the pathogenesis of a
variety of diseases ranging from cancer to neurodegeneration.
The antioxidant response element (ARE) signaling pathway is
important in the amelioration of oxidative stress. Is this
molecule agonists of antioxidant response element (ARE)
signaling pathway? [MASK]"
```

For Galactica, the answer is expected to be generated after reading the question. The input example is

```
"[START_I_SMILES] CCOc1ccccc1-n1nnnc1SCC(=O)NC(=O)NCc1ccco1
[END_I_SMILES]
Question: Oxidative stress has been implicated in the
pathogenesis of a variety of diseases ranging from cancer to
neurodegeneration. The antioxidant response element (ARE)
signaling pathway is important in the amelioration of
oxidative stress. Is this molecule agonists of antioxidant
response element (ARE) signaling pathway?
Answer:"
```

For MoMu, we compute the matching score between the molecule graph and the instruction with each answer. In the example, the classification scores for 'Yes' and 'No' are computed by matching graph feature of molecule CCOc1ccccc1-n1nnnc1SCC(=O)NC(=O)NCc1ccco1 with

```
"Oxidative stress has been implicated in the pathogenesis of a
variety of diseases ranging from cancer to neurodegeneration.
The antioxidant response element (ARE) signaling pathway is
important in the amelioration of oxidative stress. Is this
molecule agonists of antioxidant response element (ARE)
signaling pathway? Yes"
```

and

"Oxidative stress has been implicated in the pathogenesis of a variety of diseases ranging from cancer to neurodegeneration. The antioxidant response element (ARE) signaling pathway is important in the amelioration of oxidative stress. Is this molecule agonists of antioxidant response element (ARE) signaling pathway? No"

.

# B Method

## B.1 Discussion of Individual Encoding Module Method

The individual encoding module-based multimodal language model can be formalized as $\text{LLM}(M(G), T)$, where $M$ is the individual encoding module for graph data $G$. For example, the visual module is applied to pre-encode the image data to get the dense representation, then put into the language model as tokens embedding [4, 9, 1, 32]. Current works on molecule language models also use a GNN to get the representation of molecules to interact with the language models [17, 59, 56].

This method can be considered as decomposition of the conditional probability $P(\hat{y}|G, T)$

$$P(\hat{y}|G, T) = \int P_M(z|G) P_{\text{LLM}}(\hat{y}|z, T) dz, \tag{6}$$

based on the assumption that the feature distributions $P(z|G)$ should be modeled by modality-specific modules to introduce inductive bias, and be independent of text information to help with adaptation to novel text data.

However, for the molecule-text model, individual pre-encoding modules present problems. First, graph learning relies on structure information, but the dense vectors encoded by GNN have a limited capacity to carry structure information, and language models don't have inductive bias toward graph structure. Furthermore, training the additional module is difficult due to the increased layers, since deep transformers have vanishing gradients in early layers [34, 2], which is a well-known problem of transformer. Lastly, the additional modules increase parameters and training costs.

Our method GIMLET not only overcome these issues, our approach GIMLET not only directly unifies the standard language model for graph and text *without* introducing additional graph encoder module, but also remains the decoupled graph encoding for better generalization.

## B.2 Model Theoretical Capacity

In this section, we analyze the theoretical capacity of our modeling method.

**Theorem 1** *Assume for different input features and position embeddings, the transformer layers can output different output features. The transformer with distance-based relative position embedding has a stronger capacity than the 1-WL test for the graph isomorphism problem.*

**Proof 1** *The 1-WL test is defined as the following iteration:*

$$\begin{aligned}
\chi_G^0(i) &= \text{hash}(v_i) \\
\chi_G^t(i) &:= \text{hash}\left(\chi_G^{t-1}(i), \left\{\chi_G^{t-1}(j) : j \in \mathcal{N}_G(i)\right\}\right) (\forall i \in N),
\end{aligned} \tag{7}$$

*where $\chi_G$ is the label in WL test,* $\text{hash}$ *is the hash function,* $\mathcal{N}_G(i)$ *is the neighbor of node $i$.*

*The transformer with distance-based relative position embedding can be considered as the following mapping:*

$$
\begin{aligned}
\chi_G^t(i) &:= \mathrm{hash}\left(\left\{\left(d_G(i,j), \chi_G^{t-1}(j)\right) : j \in N\right\}\right) \\
&= \mathrm{hash}(\{(0, \chi_G^{t-1}(i))\} \\
&\quad \cup \{(1, \chi_G^{t-1}(j)) : j \in \mathcal{N}_G(i)\} \\
&\quad \cup \{(d_G(i,k), \chi_G^{t-1}(k)) : k \in N - \mathcal{N}_G(i) - \{i\}\})
\end{aligned}
\tag{8}
$$

*It can be seen that the iteration of the transformer with distance-based relative position embedding includes both the node $i$ and its neighbors $\mathcal{N}_G(i)$, marked by distance 0 and 1, respectively, ensuring the capacity is at least as strong as 1-WL test. It further includes other nodes far away, along with their distance, which constitutes a stronger capacity than 1-WL test. Figure 5 are two example graphs that cannot be distinguished by 1-WL test, but can be distinguished by transformer with distance-based relative position embedding.*

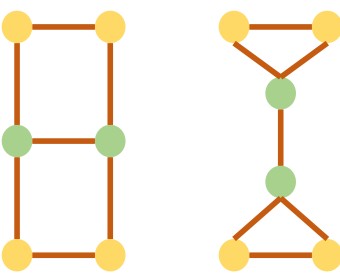

Figure 5: Two example graphs that cannot be distinguished by 1-WL test, but can be distinguished by transformer with distance-based relative position embedding.

**Theorem 2** *Assume for different input features and position embeddings, the transformer layers can output different output features.* GIMLET *can distinguish graph-instruction pairs if graphs can be distinguished by transformer with distance-based relative position embedding, or instructions are different.*

**Proof 2** *Because* GIMLET *decomposes the attention from graph nodes to text, the graph nodes can only attend to other graph nodes. Thus the encoding capacity of graph data is the same as a single transformer with distance-based relative position embedding for graph data.*

*Along with the assumption of transformer layers,* GIMLET *is able to distinguish graph-instruction pairs if graphs can be distinguished by transformer with distance-based relative position embedding, or instructions are different.*

### B.3 Detailed Related Work

We present a detailed related work here, due to the space limitation of paper.

**Molecule Representation learning** In recent years, there has been a growing interest in developing molecular representation learning for downstream tasks like drug discovery and other applications. One approach that has received considerable attention is utilizing language modeling techniques to acquire molecular representations based on Simplified Molecular Input Line Entry System (SMILES) strings [68, 10]. Although sequence-based representations have demonstrated success in some applications, concerns have been raised about their capability to incorporate all pertinent substructure information. To address this limitation, some researchers have proposed the use of Graph Neural Networks (GNNs) to model molecules as graphs [21, 83, 26], potentially providing a more comprehensive and accurate representation of the molecular structure.

Existing GNNs follow the message-passing paradigm and suffer from problems like long-range dependency vanishing and over-smoothing. Recently, Graph Transformer [52, 80] has been proposed to better encode structures of graphs. The Graph Transformer is inspired by the Transformer

architecture, which has shown remarkable performance in natural language processing [65, 13, 41]. The Graph Transformer extends the Transformer architecture to the graph domain, allowing the model to capture the global structure and long-range dependencies of the graph [86, 15, 31, 29, 74, 48, 42, 80, 8, 43, 11, 5, 23, 88].

**Molecule Pretraining** To fully explore the inherent structural information of molecules on a large scale and transfer useful information to downstream tasks, significant efforts have been made to address the inadequacies in molecular pre-training. Supervised pretraining is commonly used for learning useful representations [26, 80, 62]. As for unsupervised pretraining, one approach involved using an generative pre-training strategy on molecular SMILES strings [68, 25, 10, 3, 53] and Graph [26, 37, 52, 87], which was followed by recent works adopting the contrastive paradigm that aligns representation of augmented views of the same graph but keeping views from other graphs away [67, 60, 24, 83, 82, 63, 78, 19, 61, 70, 76, 69, 40].

The pretraining methods mentioned focus on obtaining representations for supervised training. However, for natural language instruction-based zero-shot graph learning, it's necessary to incorporate natural language into the pretraining process. Several studies have explored molecule structure-text multimodal pretraining. One class of method is the SMILES based language model, including KVPLM [85] and MolT5 [16], which use SMILES strings and text for joint representation and translation. Another work Galactica [64] explored the multi-task molecule task learning with instruction. Some other works acquire advanced representations for molecules by GNN, such as Text2Mol [17], MoMu [59], MoleculeSTM [39], and CLAMP [56], trained by contrastive learning between molecule graph and text description for molecule retrieval and caption tasks. MoleculeSTM and CLAMP explored molecule editing and property prediction with instructions. However, none of these works address the zero-shot fashion on complex molecule tasks like property prediction, due to constraints imposed by the pretraining methodology that not addressing the instruction-following ability, and their model capacity for representing molecule graphs.

**Instruction-based zero-shot learning** Instruction-based zero-shot learning is an innovative approach that leverages natural language instructions and definitions to enable neural models to solve a variety of tasks [50, 6, 55, 18, 89, 44, 45, 49]. By providing a human-readable prompt, this method enables easier and more efficient specification of the learning task by utilizing knowledge about the task without data. To enhance the model's ability to follow instructions, some researchers have employed instruction-based pretraining techniques [54, 71, 12, 47], which explicitly train language models to solve tasks with instructions. Besides natural language processing, instruction-based zero-shot learning is also studied in multimodal domains like images [4, 9, 1, 32].

# C  Experiments

## C.1  Experiment setting

Our model only utilizes the basic features [26, 62] of molecule graphs, which do not include additional features like ring markers. Specifically, it utilizes the first two dimensions of node features and the first two dimensions of edge features processed by ogb.smiles2graph. Therefore, the effectiveness of GIMLET predominantly stems from its architectural design and pretraining rather than the graph features it incorporates.

Following the standard supervised setting in previous studies [26], we utilize the scaffold strategy [51] to partition datasets into three subsets: the training set, validation set, and testing set with a ratio of 0.8, 0.1, 0.1. The scaffold strategy is a deterministic approach that involves sorting the data based on the scaffold, which represents the molecular structure. While this strategy aids in dataset partitioning, it can introduce a significant domain gap between the training and testing sets, thereby increasing the challenge of generalization.

For zero-shot, we report the results on the testing sets, ensuring the comparability of our results to previous works. For few-shot, we report the result of the best validation model on the testing set, the same as previous works and other supervised baselines [51].

Many datasets encompass multiple tasks. To evaluate these datasets, we conduct separate testing for each task, accompanied by their respective instructions. For datasets with multiple tasks, we report the average ROC-AUC score for each task, following the methodology established in previous works [26].

## C.2  Detailed Zero-Shot Result

We list the full zero-shot result of GIMLET and baselines in Table 10, 11, and 12. The standard deviation for supervised results are denoted after ±, and the multi-task setting results of Galactica are denoted in parentheses with italic. We also include the instruction-based zero-shot result reported in recent baseline CLAMP [56] which is tested by their instruction, denoted by italics too. CLAMP is a contrastive pretrained model with ensembled encoders for molecule and text. The parameter number for CLAMP's result is not clearly stated in their paper but should be larger than 10B as they use sT5 language model [50] XXL variant (11B) as one of the ensembled language models.

Table 10: Zero shot performance over Bio-activity tasks

| Method | # Parameter | Type | bace | hiv | muv | Avg. bio |
|---|---|---|---|---|---|---|
| KVPLM | 110M | | 0.5126 | 0.6120 | 0.6172 | 0.5806 |
| MoMu | 113M | Zero Shot | 0.6656 | 0.5026 | 0.6051 | 0.5911 |
| CLAMP | > 10B | | *0.6476* | *0.8067* | - | - |
| GIMLET | 64M | | 0.6957 | 0.6624 | 0.6439 | 0.6673 |
| Galactica-125M | 125M | Multi Task | 0.4451(*0.561*) | 0.3671(*0.702*) | 0.4986 | 0.4369 |
| Galactica-1.3B | 1.3B | | 0.5648(*0.576*) | 0.3385(*0.724*) | 0.5715 | 0.4916 |
| GCN | 0.5M | | *0.736±0.030* | *0.757±0.011* | *0.732±0.014* | 0.742 |
| GAT | 1.0M | | *0.697±0.064* | *0.729±0.018* | *0.666±0.022* | 0.697 |
| GIN | 1.8M | Supervised | *0.701±0.054* | *0.753±0.019* | *0.718±0.025* | 0.724 |
| Graphormer | 48M | | 0.7760±0.015 | 0.7452±0.014 | 0.7061±0.027 | 0.7424 |
| Graphormer-p | 48M | | 0.8575±0.006 | 0.7788±0.012 | 0.7480±0.020 | 0.7948 |

Table 11: Zero shot performance over Toxicity tasks

| Method | # Parameter | Type | tox21 | toxcast | Avg. tox |
|---|---|---|---|---|---|
| KVPLM | 110M | | 0.4917 | 0.5096 | 0.5007 |
| MoMu | 113M | Zero Shot | 0.5757 | 0.5238 | 0.5498 |
| CLAMP | > 10B | | *0.6058* | *0.5383* | 0.5721 |
| GIMLET | 64M | | 0.6119 | 0.5904 | 0.6011 |
| Galactica-125M | 125M | Multi Task | 0.4964(*0.543*) | 0.5106(*0.518*) | 0.5035 |
| Galactica-1.3B | 1.3B | | 0.4946(*0.606*) | 0.5123(*0.589*) | 0.5035 |
| GCN | 0.5M | | *0.749±0.008* | *0.633±0.009* | 0.691 |
| GAT | 1.0M | | *0.754±0.005* | *0.646±0.006* | 0.700 |
| GIN | 1.8M | Supervised | *0.740±0.008* | *0.634±0.006* | 0.687 |
| Graphormer | 48M | | 0.7589±0.004 | 0.6470±0.008 | 0.7029 |
| Graphormer-p | 48M | | 0.7729±0.006 | 0.6649±0.006 | 0.7189 |

Table 12: Zero shot performance over Pharmacokinetic tasks

| Method | # Parameter | Type | bbbp | cyp450 | Avg. pha |
|---|---|---|---|---|---|
| KVPLM | 110M | | 0.6020 | 0.5922 | 0.5971 |
| MoMu | 113M | Zero Shot | 0.4981 | 0.5798 | 0.5390 |
| CLAMP | > 10B | | *0.4788* | - | - |
| GIMLET | 64M | | 0.5939 | 0.7125 | 0.6532 |
| Galactica-125M | 125M | Multi Task | 0.6052(*0.393*) | 0.5369 | 0.5711 |
| Galactica-1.3B | 1.3B | | 0.5394(*0.604*) | 0.4686 | 0.5040 |
| GCN | 0.5M | | *0.649±0.030* | *0.8041±0.005* | 0.7266 |
| GAT | 1.0M | | *0.662±0.026* | 0.8281±0.004 | 0.7451 |
| GIN | 1.8M | Supervised | *0.658±0.045* | 0.8205±0.012 | 0.7392 |
| Graphormer | 48M | | 0.7015±0.013 | 0.8436±0.003 | 0.7725 |
| Graphormer-p | 48M | | 0.7163±0.009 | 0.8877±0.004 | 0.8020 |

The result in parentheses represents the outcome of the multitask setting, also referred to as weakly supervised in the original paper, where the same instructions are used for both pretraining and testing. While Galactica has been exposed to the same task instructions, it actually employs multitask learning with instructions serving as task identity.

Even in comparison to Galactica's multitask result, GIMLET demonstrates comparable or superior performance on most datasets. This highlights the ability of GIMLET to perform zero-shot tasks with high quality.

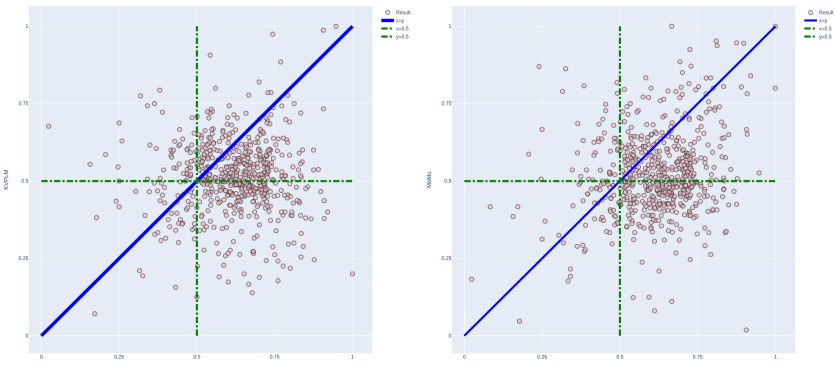

Figure 6: Scatter of GIMLET over baselines. Below the diagonal line x=y means our method performs better.

The disparity between the multitask result and the tested result with our instructions is due to the gap between their instructions and ours, which indicates that Galactica relies on specific task instructions for task recognition, without a true understanding of the instructions. As a result, it exhibits poor generalization to other instruction forms. Note that Galactica even do not surpass KVPLM and MoMu which are also zero-shot learning methods.

GIMLET exhibits superior performance compared to the larger model CLAMP on the majority of datasets, with the exception of HIV. It is important to highlight that our model is significantly smaller in size than CLAMP, underscoring the effectiveness of our unified graph-text language model. Additionally, it should be noted that CLAMP lacks the capability to handle regression tasks due to its contrastive model architecture, whereas our encoder-decoder architecture enables us to successfully tackle a wide range of task types.

Significantly, the supervised results shed light on the task difficulties associated with each dataset. This showcases GIMLET's capability to effectively solve molecule tasks in a zero-shot manner, approaching the performance of supervised results. Furthermore, our pretraining tasks yield an average performance improvement of 3 percent for Graphormer, with the largest gains observed in Bioactivity tasks and the smallest in Toxicity tasks. This suggests that there still exist gaps between the pretraining data and our downstream tasks, addressing the zero-shot setting of our dataset.

In Figure 6, we present scatter plots comparing GIMLET with KVPLM and MoMu across all tasks. The diagonal line represents the equality line where x=y indicates our method outperforms the baseline. Notably, it is evident that GIMLET consistently performs significantly better than random guessing and surpasses the baselines on all tasks.

We plot the scatter of regression tasks in Figure 7. The plot clearly demonstrates a strong correlation between the predicted and actual values for ESOL and Lipo.

### C.3 Detailed Few-Shot Results

In both classification tasks and regression tasks, we fine-tune the last linear layer of all models using their respective modeling loss.

It is important to note that the instruction-based few-shot approach is trained on each task individually, while supervised baselines are trained on multiple tasks from the dataset. Therefore, comparing these two approaches may not be strictly fair, as the multitask learning of the supervised baseline can contribute to improved task performance.

The results for few-shot learning on each dataset are presented in Figure 8. It is evident that, across the majority of datasets, GIMLET demonstrates improvement as the number of few-shot examples increases. In fact, it even outperforms or matches the performance of the supervised GIN on several

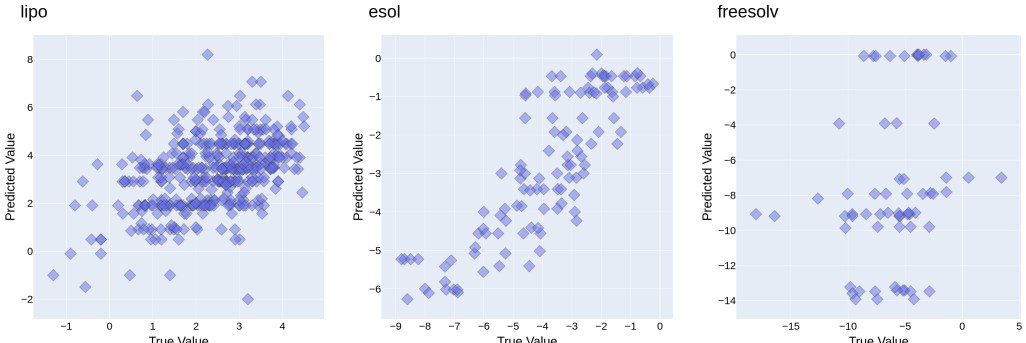

Figure 7: Scatter of GIMLET on generative tasks.

datasets, such as bace, bbbp, and esol. There is also observable enhancement in performance across various datasets when employing few-shot learning, including tox21, toxcast, lipo, and freesolv.

There is not result of MoMu on regression tasks, because MoMu is a contrastive model between graph and text, which cannot handle regression tasks.

## C.4  Detailed Ablation Results of Pretraining

The results of pretraining ablation for each dataset are presented in Table 13, 14, 15, and 16. The findings indicate that both bioactivity assay and physico-chemical properties offer significant benefits for all the downstream tasks, demonstrating positive transfer across different domains.

Table 13: Pretraining ablation study on Bio-activity tasks

|  | bace | hiv | muv | Average_bio |
|---|---|---|---|---|
| bioactivity assay only | 0.6390 | 0.6772 | 0.6044 | 0.6402 |
| physico-chemical only | 0.4648 | 0.5461 | 0.4572 | 0.4894 |
| both | 0.6957 | 0.6624 | 0.6439 | 0.6673 |

Table 14: Pretraining ablation study on Toxicity tasks

|  | tox21 | toxcast | Average_tox |
|---|---|---|---|
| bioactivity assay only | 0.5726 | 0.5625 | 0.5676 |
| physico-chemical only | 0.4478 | 0.5017 | 0.4748 |
| both | 0.6119 | 0.5904 | 0.6011 |

## C.5  Instruction Robustness

To test the robustness of GIMLET, the Instructions are rephrased by GPT-3.5-turbo. There are four types of rephrasing, realized by the following prompts:

rewrite

```
'Rephrase the text  of the following prompt: \n'
```

expand

```
'Rephrase the text  of the following prompt longer: \n'
```

detail

```
'Rephrase the text  of the following prompt by adding more
explanation: \n'
```

short

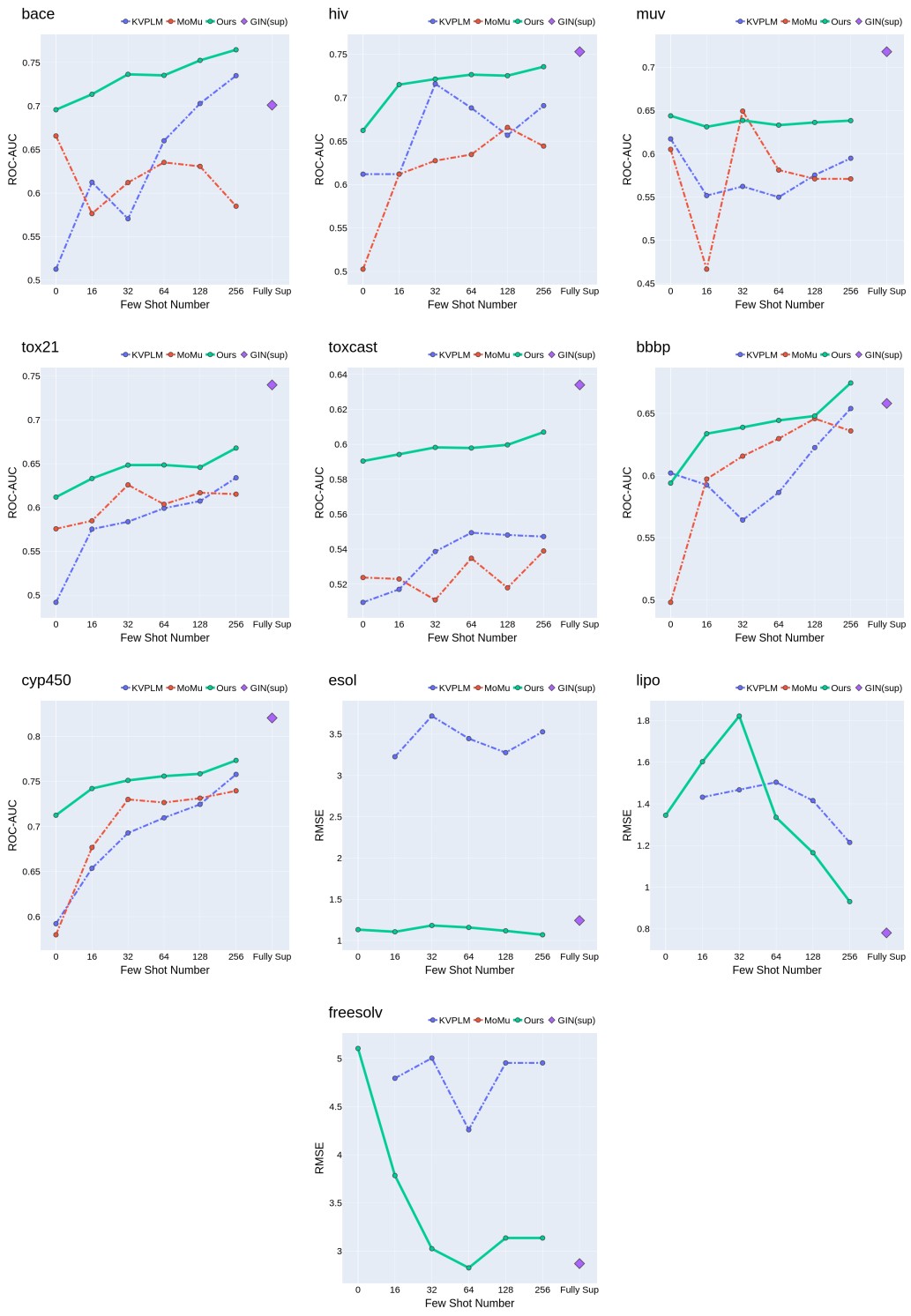

Figure 8: Few-shot performance on each dataset

Table 15: Pretraining ablation study on Pharmacokinetic tasks

|  | bbbp | cyp450 | Average_pha |
| --- | --- | --- | --- |
| bioactivity assay only | 0.5313 | 0.6829 | 0.6071 |
| physico-chemical only | 0.5932 | 0.4976 | 0.5454 |
| both | 0.5939 | 0.7125 | 0.6532 |

Table 16: Pretraining ablation study on Physical-chemical tasks

| | esol | freesolv | lipo | Average_phy |
|---|---|---|---|---|
| bioactivity assay only | - | - | - | |
| physico-chemical only | 1.1822 | 5.2935 | 1.3778 | 2.6178 |
| both | 1.1320 | 5.1027 | 1.3450 | 2.5266 |

```
'Rephrase the text  of the following prompt shorter: \n'
```

Given a task instruction, we rephrase the instruction by the prompts above. Here is an example of four types of rephrased task instruction from Toxcast:

origin

```
"CEETOX_H295R_ANDR, is one of 23 assay component(s) measured
or calculated from the CEETOX_H295R assay. It is designed to
make measurements of hormone induction, a form of inducible
reporter, as detected with absorbance signals by HPLC-MS-MS
technology.Data from the assay component CEETOX_H295R_ANDR was
analyzed into 2 assay endpoints. This assay endpoint,
CEETOX_H295R_ANDR_dn, was analyzed in the positive fitting
direction relative to DMSO as the negative control and
baseline of activity. Using a type of inducible reporter,
loss-of-signal activity using HPLC-MS-MS was used to
understand synthesis of Androstenedione in H295R cell line at
48hr of chemical exposure. To generalize the intended target
to other relatable targets, this assay endpoint is annotated
to the steroid hormone intended target family, where the
subfamily is androgens. Is this molecule effective to this
assay?"
```

rewrite

```
"The CEETOX_H295R_ANDR is a component of the CEETOX_H295R
assay, which measures hormone induction through absorbance
signals detected by HPLC-MS-MS technology. The data from
CEETOX_H295R_ANDR has been analyzed into two endpoints, with
CEETOX_H295R_ANDR_dn being analyzed in relation to DMSO as the
negative control and baseline of activity. HPLC-MS-MS was used
to detect loss-of-signal activity and understand the synthesis
of Androstenedione in H295R cell line after 48 hours of
chemical exposure. This assay endpoint is related to the
steroid hormone intended target family, specifically the
subfamily of androgens, and can be generalized to other
similar targets.Can this assay be effectively performed using
this molecule?"
```

expand

```
"The CEETOX_H295R_ANDR assay component is just one of the 23
assay components that are measured or calculated from the
CEETOX_H295R assay. The assay is specifically designed to
measure hormone induction, which is a form of inducible
reporter, and is detected using absorbance signals by means of
HPLC-MS-MS technology. The data obtained from the
CEETOX_H295R_ANDR assay component was analyzed into two assay
endpoints. The CEETOX_H295R_ANDR_dn assay endpoint was
analyzed in the positive fitting direction in relation to DMSO
as the negative control and activity baseline. To understand
the synthesis of Androstenedione in the H295R cell line after
48 hours of chemical exposure, loss-of-signal activity was
```

```
used with HPLC-MS-MS technology. This endpoint is annotated to
the steroid hormone intended target family to help other
related targets, where the subfamily is androgens. Can it be
determined if this particular molecule exhibits desirable
efficacy to be utilized in this particular assay?"
```

detail

```
"The CEETOX_H295R_ANDR is an assay component that is one of
the 23 components that are measured or calculated from the
CEETOX_H295R assay. It is intended to measure hormone
induction, which is a form of inducible reporter, and the
measurement is done with the help of absorbance signals using
HPLC-MS-MS technology. The data obtained from the measurement
of assay component CEETOX_H295R_ANDR is analyzed into two
assay endpoints. One of these endpoints, CEETOX_H295R_ANDR_dn,
is analyzed in the positive fitting direction, relative to
DMSO, which is used as the negative control and baseline for
activity. The HPLC-MS-MS technology is used to detect the
loss-of-signal activity, which helps in understanding the
synthesis of Androstenedione in H295R cell line after 48 hours
of chemical exposure. To make the intended target more
comprehensive and relatable to other targets, the assay
endpoint is annotated to the steroid hormone intended target
family, where the subfamily is androgens. Can this molecule be
used for this assay?"
```

short

```
"CEETOX_H295R_ANDR is one of 23 components in the CEETOX_H295R
assay, measuring hormone induction detected with absorbance
signals by HPLC-MS-MS. It's analyzed into 2 endpoints, with
CEETOX_H295R_ANDR_dn being the positive fitting direction
relative to the negative control. It analyzes the
loss-of-signal activity to understand Androstenedione
synthesis in H295R cell line after 48hr chemical exposure.
It's annotated as a steroid hormone intended target in
androgens sub-family. Is molecule suitable for assay?"
```

## C.6  Instruction Ablation

To ablate the explanation-based instruction, we remove the explanation and only keep the assay name. The ablated instruction for the instruction above is:

```
"The assay name is CEETOX_H295R_ANDR. Is this molecule
effective to this assay?"
```

## C.7  Attention Visualization

We present visualizations of the attention of text tokens to molecule graphs, demonstrating how our unified transformer incorporates molecule information using various instructions. We randomly sample molecules and attention heads for visualization. To emphasize high-level features, we focus on visualizing the attention patterns of the last layer. The redder means the larger attention value.

For BACE instruction, we visualize the attention of several keywords marked in red to molecules:

"BACE1 is an aspartic-acid protease important in the pathogenesis of Alzheimer's disease, and in the formation of myelin sheaths. BACE1 is a member of family of aspartic proteases. Same as other aspartic proteases, BACE1 is a bilobal enzyme, each lobe contributing a catalytic Asp residue, with an extended active site cleft localized between the two lobes of the molecule. The assay tests whether the molecule can bind to the BACE1 protein. Is this molecule effective to the assay?"

For BBBP instruction:

'In general, molecules that passively diffuse across the brain blood barrier have the molecular weight less than 500, with a LogP of 2-4, and no more than five hydrogen bond donors or acceptors. Does the molecule adhere to the three rules or not?'

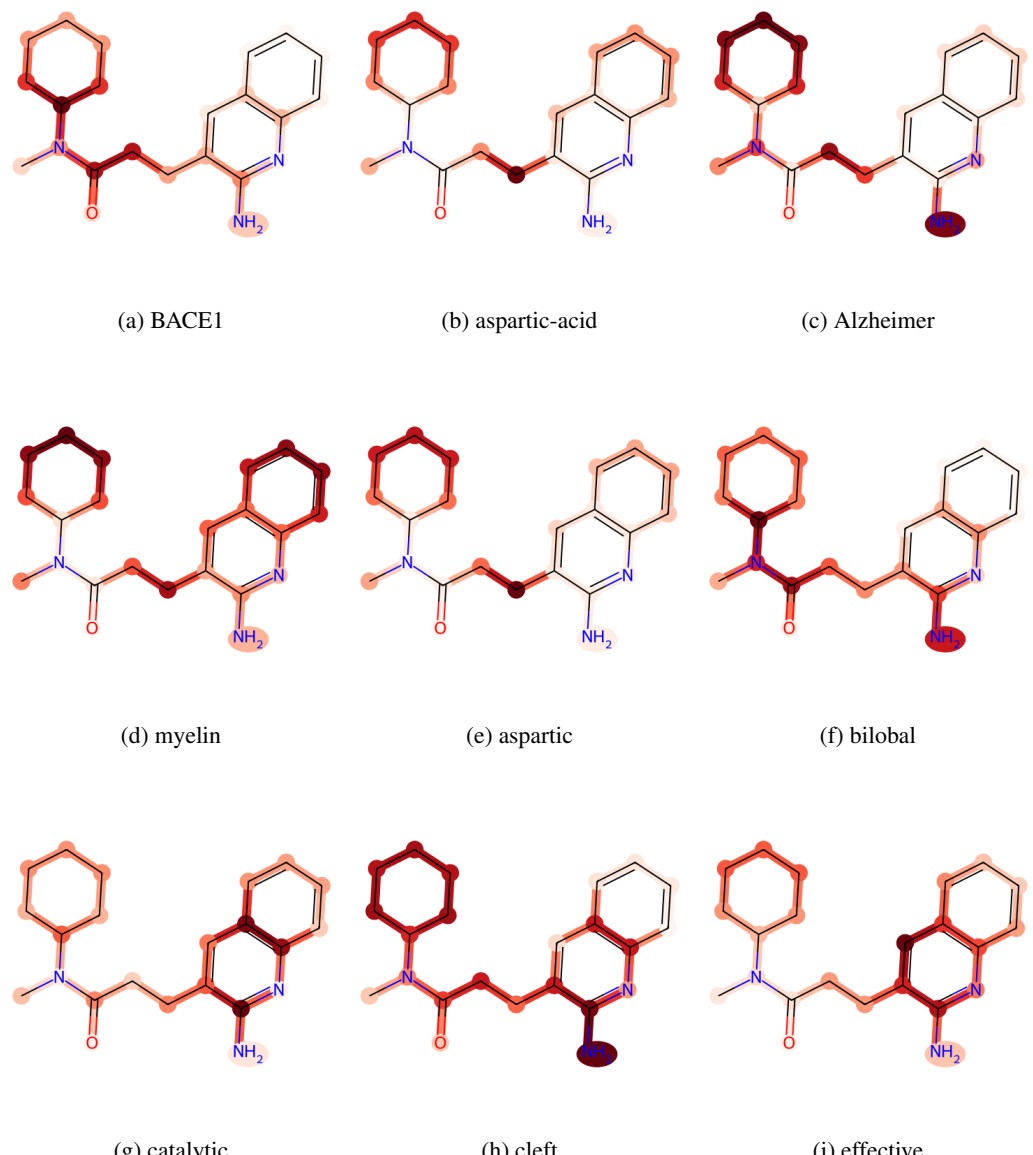

(a) BACE1     (b) aspartic-acid     (c) Alzheimer

(d) myelin     (e) aspartic     (f) bilobal

(g) catalytic     (h) cleft     (i) effective

Figure 9: Visualization of attention for BACE on molecule
O=C(N(C)C1CCCCC1)CCc1cc2c(nc1N)cccc2

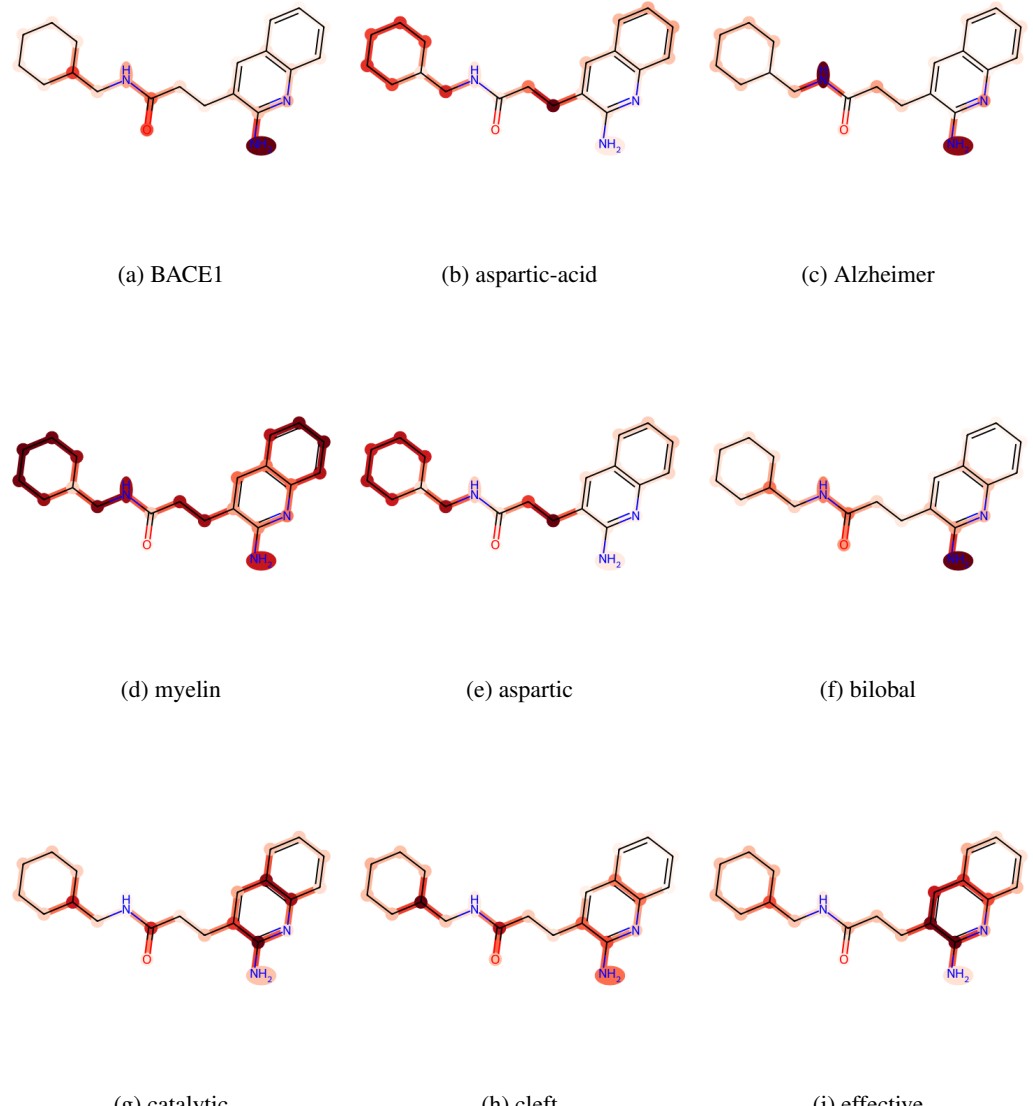

(a) BACE1

(b) aspartic-acid

(c) Alzheimer

(d) myelin

(e) aspartic

(f) bilobal

(g) catalytic

(h) cleft

(i) effective

Figure 10: Visualization of attention for BACE on molecule
O=C(NCC1CCCCC1)CCc1cc2c(nc1N)cccc2

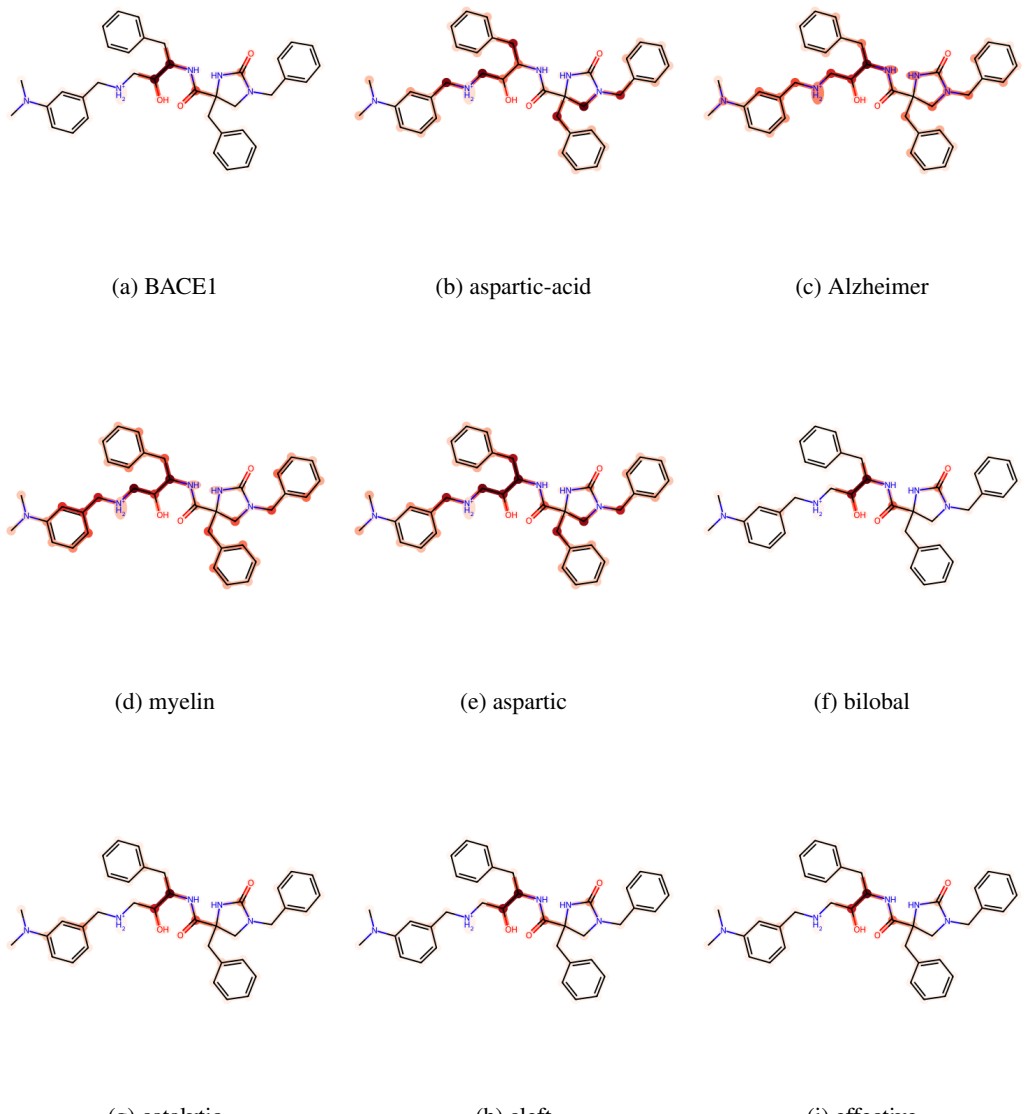

(a) BACE1             (b) aspartic-acid             (c) Alzheimer

(d) myelin             (e) aspartic             (f) bilobal

(g) catalytic             (h) cleft             (i) effective

Figure 11: Visualization of attention for BACE on molecule
O=C1NC(CN1Cc1ccccc1)(Cc1ccccc1)C(=O)NC(Cc1ccccc1)C(O)C[NH2+]Cc1cc(N(C)C)ccc1

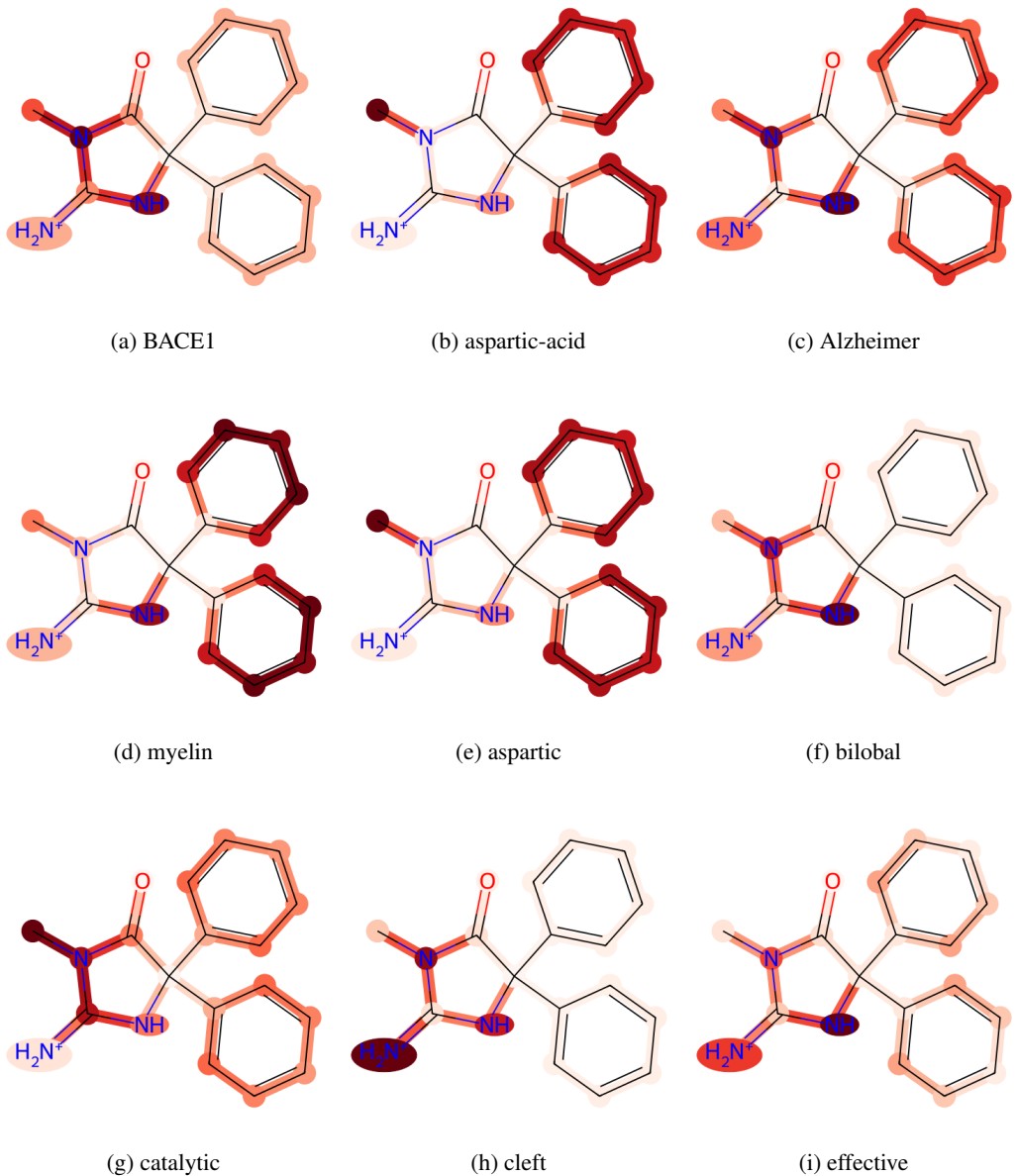

(a) BACE1        (b) aspartic-acid        (c) Alzheimer

(d) myelin        (e) aspartic        (f) bilobal

(g) catalytic        (h) cleft        (i) effective

Figure 12: Visualization of attention for BACE on molecule
O=C1N(C)C(=[NH2+])NC1(c1ccccc1)c1ccccc1

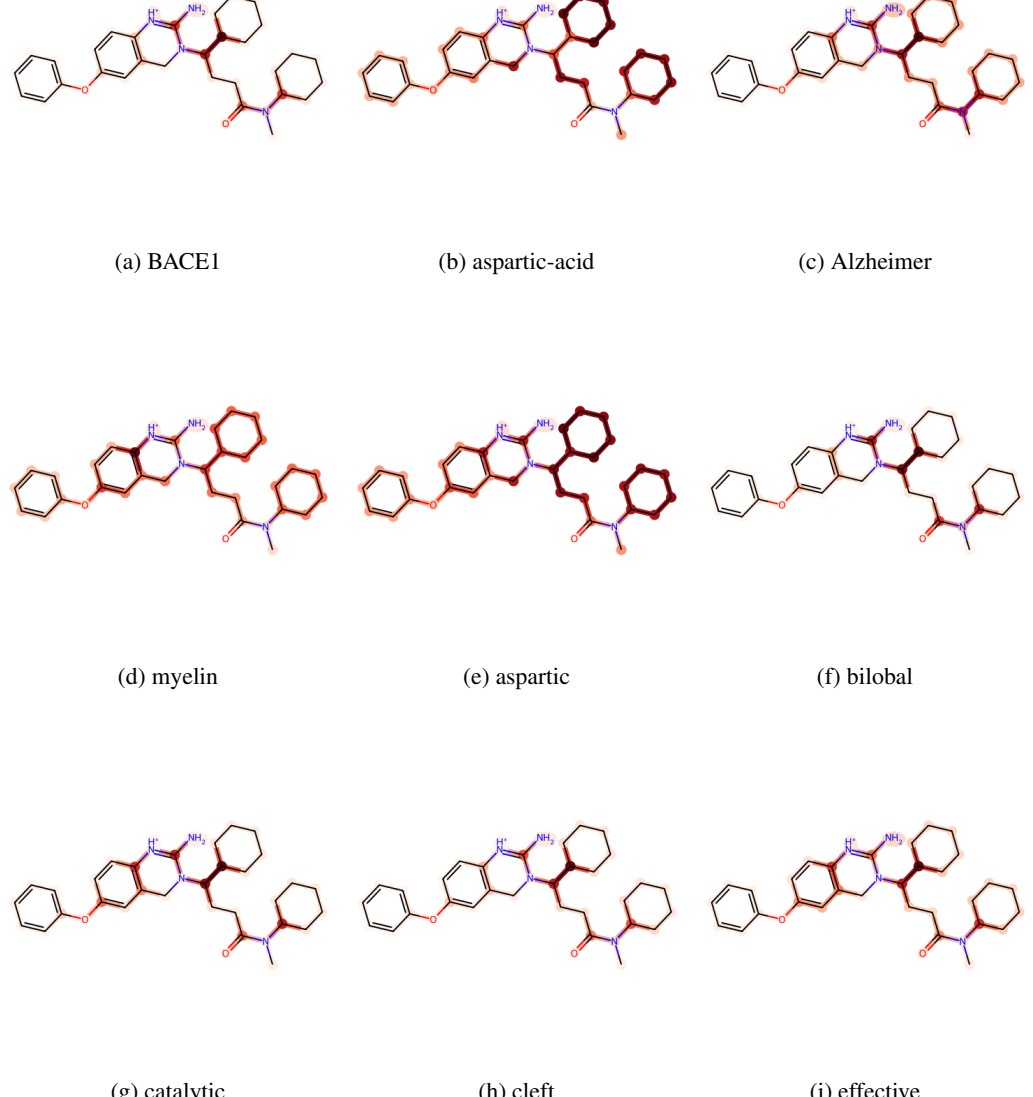

(a) BACE1        (b) aspartic-acid        (c) Alzheimer

(d) myelin        (e) aspartic        (f) bilobal

(g) catalytic        (h) cleft        (i) effective

Figure 13: Visualization of attention for BACE on molecule
O(c1cc2CN(C(CCC(=O)N(C)C3CCCCC3)C3CCCCC3)C(=[NH+]c2cc1)N)c1ccccc1

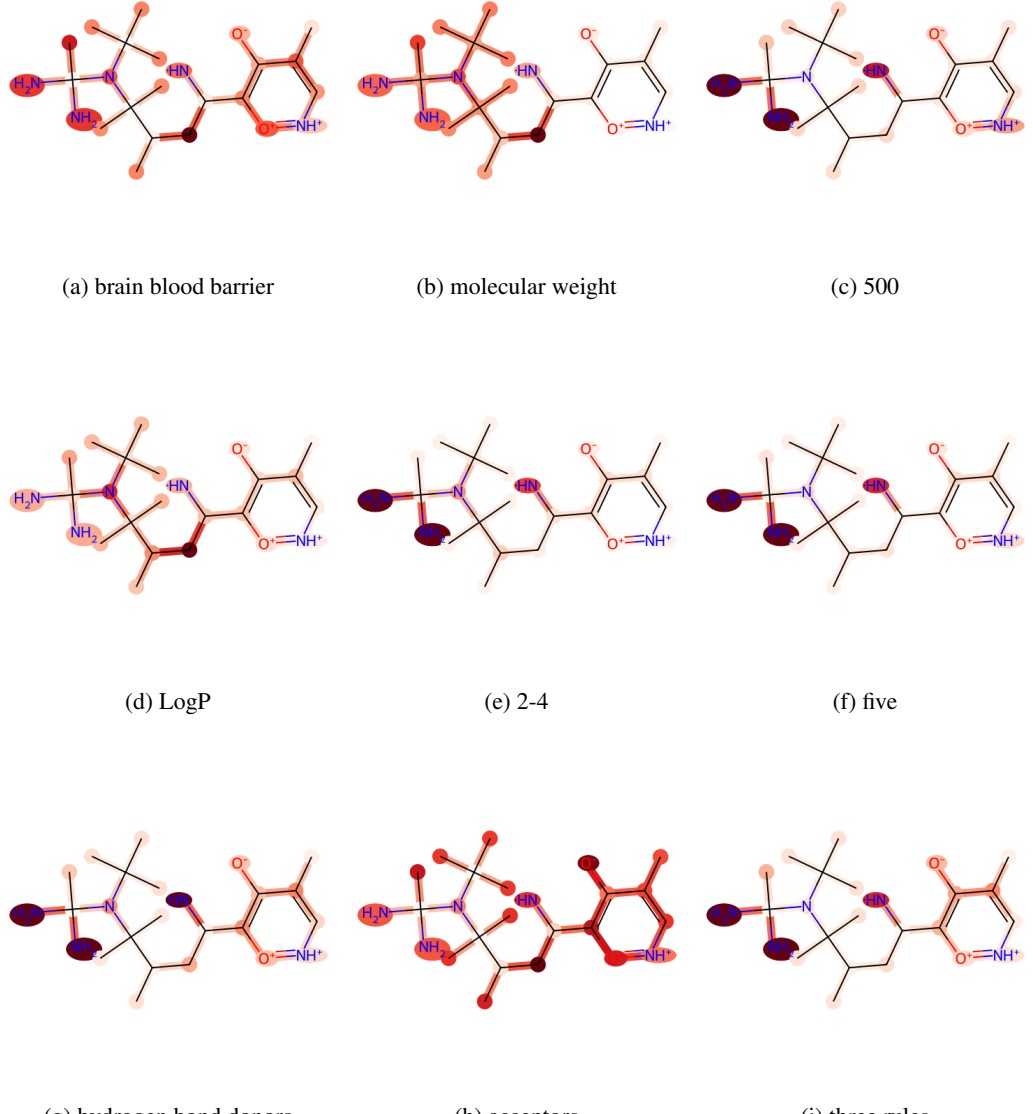

(a) brain blood barrier      (b) molecular weight      (c) 500

(d) LogP      (e) 2-4      (f) five

(g) hydrogen bond donors      (h) acceptors      (i) three rules

Figure 14: Visualization of attention for BBBP on molecule
[NH]C(CC(C)C([N@@](C(C)(C)C)C(N)(C)N)(C)C)c1c(c(c[nH+][o+]1)C)[O-]

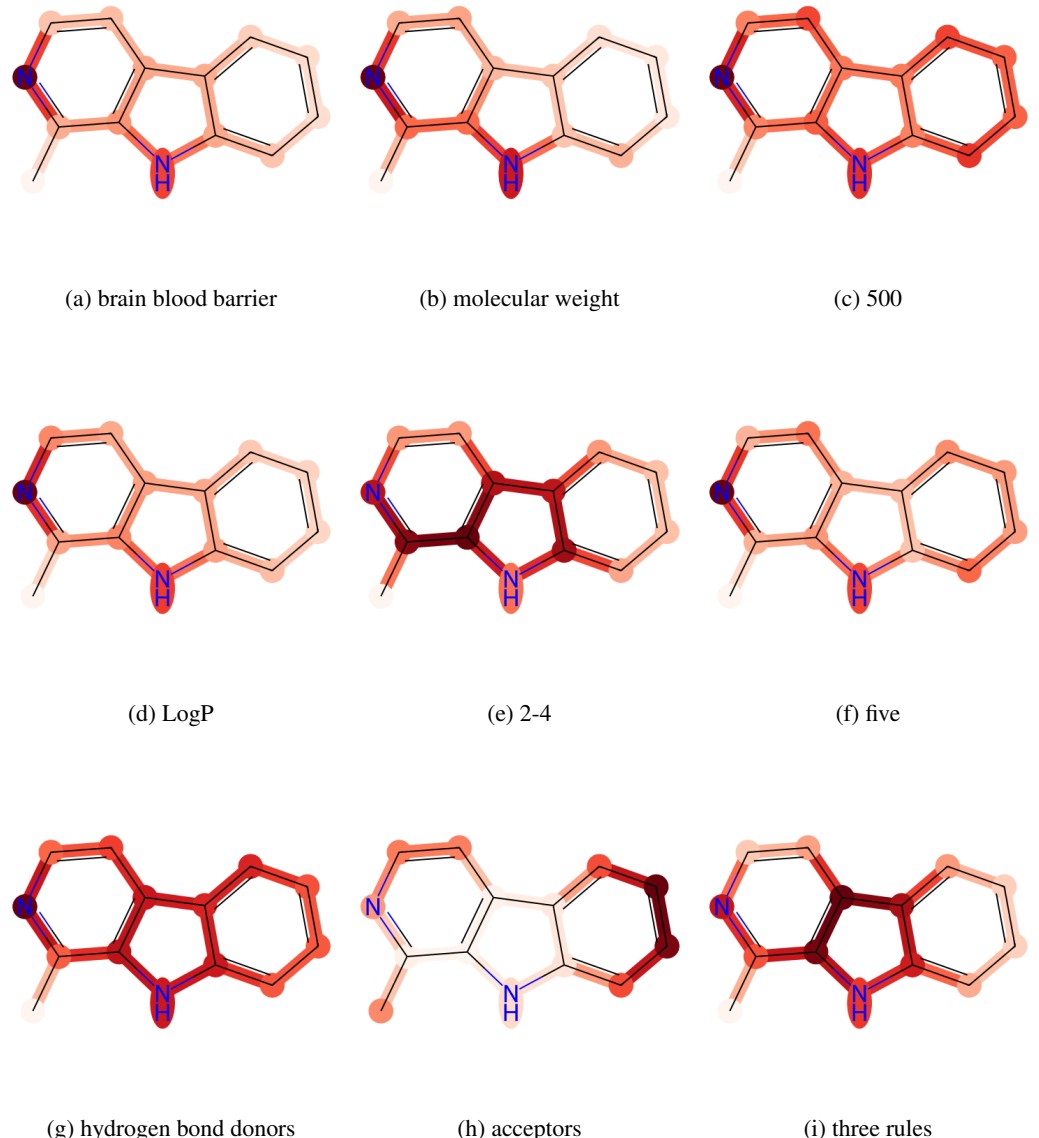

(a) brain blood barrier

(b) molecular weight

(c) 500

(d) LogP

(e) 2-4

(f) five

(g) hydrogen bond donors

(h) acceptors

(i) three rules

Figure 15: Visualization of attention for BBBP on molecule
Cc1nccc2c1[nH]c3ccccc23

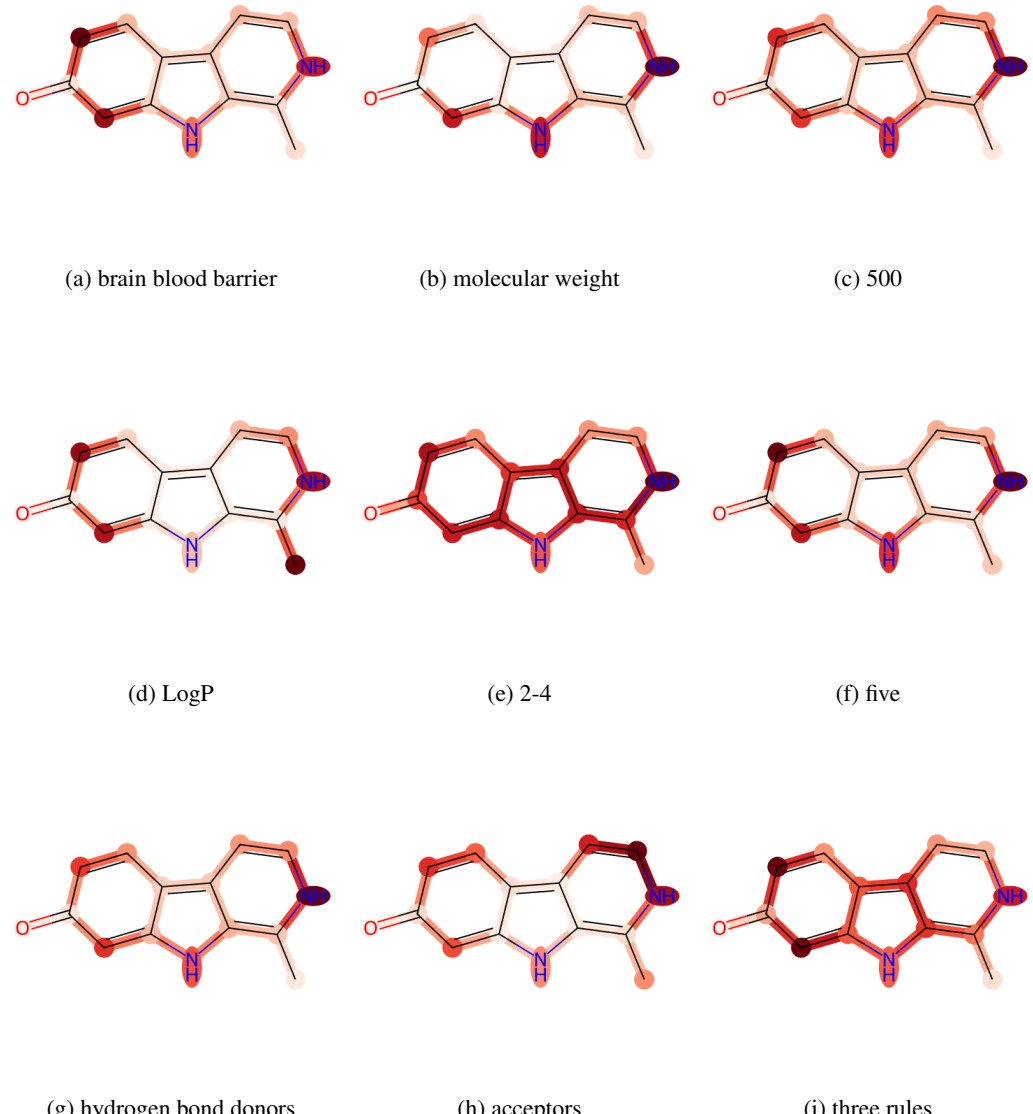

(a) brain blood barrier  (b) molecular weight  (c) 500

(d) LogP  (e) 2-4  (f) five

(g) hydrogen bond donors  (h) acceptors  (i) three rules

Figure 16: Visualization of attention for BBBP on molecule
CC1=C2NC3=CC(=O)C=CC3=C2C=CN1

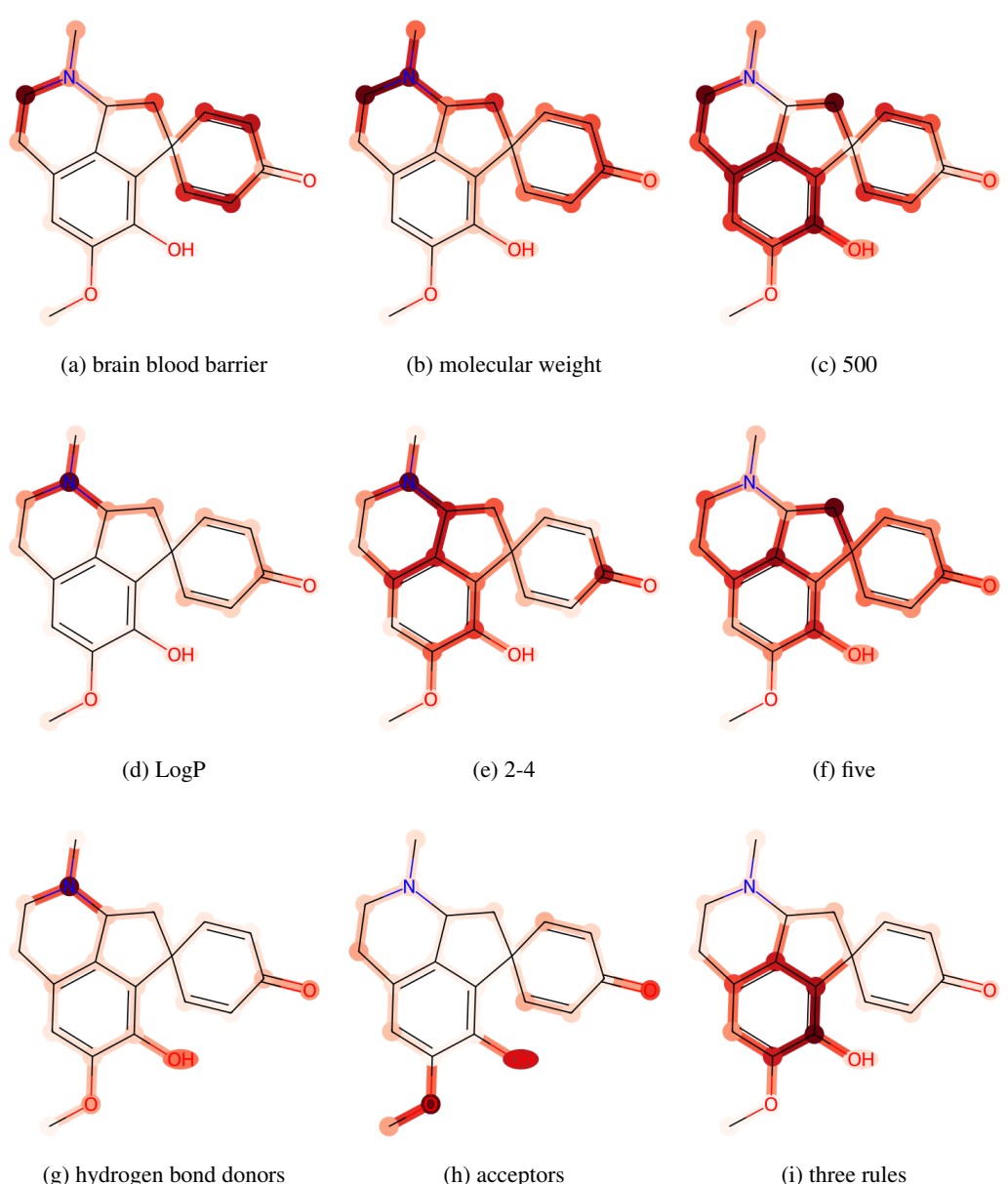

Figure 17: Visualization of attention for BBBP on molecule
COc1cc2CCN(C)C3CC4(C=CC(=O)C=C4)c(c1O)c23

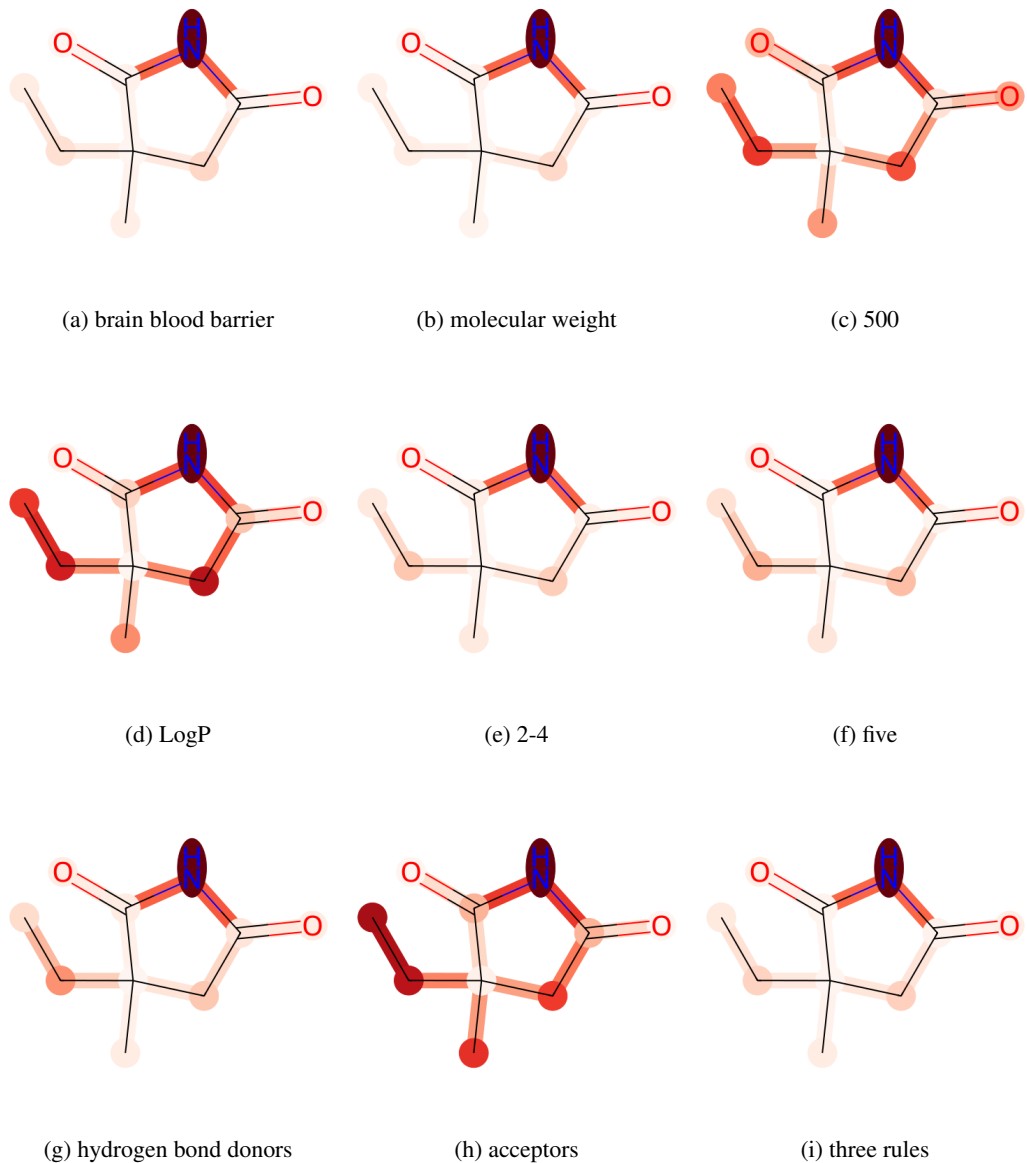

(a) brain blood barrier

(b) molecular weight

(c) 500

(d) LogP

(e) 2-4

(f) five

(g) hydrogen bond donors

(h) acceptors

(i) three rules

Figure 18: Visualization of attention for BBBP on molecule
CCC1(C)CC(=O)NC1=O

