# OpenReview forum: "GIMLET: A Unified Graph-Text Model for Instruction-Based Molecule Zero-Shot Learning"
_NeurIPS.cc/2023/Conference — NeurIPS 2023 poster_

### Official Review · Reviewer_3WHb · 2023-06-24

**Soundness:** 4 excellent
**Presentation:** 3 good
**Contribution:** 4 excellent
**Rating:** 7
**Confidence:** 4

**Summary:**

This paper introduces GIMLET, a unified graph-text model for instruction-based molecule task pretraining. It leverages natural language instructions and decoupled graph encoding techniques to improve the interaction of graphs and texts. Through experiments and evaluations, the paper demonstrates the effectiveness and robustness of GIMLET in various downstream tasks, showcasing its potential for molecule zero-shot learning.

**Strengths:**

- This paper proposed GIMLET, a novel unified graph-text model that leverages natural language instructions for pretraining. It takes each node as a token, alleviating the problem of losing structural information using separate graph encoding.
- The authors collected a set of datasets for pretraining, which can be a useful resource for the community.
- The empirical results look promising compared to previous methods.
- The paper is overall well-written and easy to follow.

**Weaknesses:**

- The authors didn't present the training details, e.g. total iterations, hyper-parameters, etc.
- GIMLET was trained from scratch with a large dataset. What are the computational requirements (GPU/memory/time) for training such a model? It seems the training can be quite expensive, especially considering the graph encoding part.


Minor:
- Page 4, line 126: need to explain what is $o_m$.
- Page 9, line 333: nature language -> natural language.

**Questions:**

- For the pretraining dataset, I understand the authors isolated the tasks for training/testing, but are the molecules also isolated for train/test, i.e. will the same molecule appear in both the training and testing set?
- For graph, node embeddings are used as input tokens. If the number of nodes varies a lot for different graphs, how do the authors handle the computational efficiency as a max length (padding) will be set for the input?

**Limitations:**

I'm not an expert in biology/chemistry, so I'm not sure if the procedures for constructing the dataset are correct.

---

> ### Author Rebuttal · Authors · 2023-08-09
>
> ### Response to question 1 in weaknesses:
>
> Thank you for your reminder regarding reproducibility. We will open-source both our pretraining dataset and code to ensure that reproducibility is achievable. We also introduce some important hyperparameters here:
>
> | Hyperparameters    | Value    |
> | ------------------ | -------- |
> | sample number      | 23.8M    |
> | batch size         | 64       |
> | epoch              | 1        |
> | total steps        | 370K     |
> | learning_rate      | 5.00E-05 |
> | dropout_rate       | 0.1      |
> | dense_act_fn       | relu     |
> | layer_norm_epsilon | 1.00E-06 |
>
> we will include these hyperparameters in the future version of the paper.
>
> ### Response to question 2 in weaknesses:
>
> Sorry for the confusion. We continue pretrain the existing T5 parameters rather than from scratch. This approach allows us to leverage the strong text understanding capabilities of the pretrained T5 model, resulting in better instruction following.
>
> The pretraining process is relatively efficient in terms of computational resources. It requires approximately 1.5 GPU days on a single V100 32G. Through techniques like gradient accumulation, it can also be performed on GPUs with smaller memory, albeit with slightly extended training times.
>
> The computation cost for graph encoding is not high, as GIMLET utilizes the same computational framework as the standard T5 model. Specifically, the graph-text mixture positional embeddings incur costs equivalent to the normal T5 relative positional embeddings.
>
> ### Response to question 3 in weaknesses:
>
> Thanks for pointing out the omission！ The $[o_1,\dots,o_m]$ denotes tokens in the instruction text, where each $o$ represents a token, and $m$ represents the length of the instruction text. We will correct these omissions in the next version of the paper.
>
> ### Response to question 1 in Questions:
>
> This is an indeed interesting question! To verify the transferability of our model to both novel tasks and new data, we split the Chembl dataset with both isolated tasks and molecules, i.e., the Chembl Zero-Shot, as the pretraining and downstream testing dataset. The experiment in Table 2 demonstrates the remarkable capability of GIMLET to transfer to Chembl Zero-Shot well, thereby validating our approach of instruction-based molecule zero-shot learning for both novel tasks and new molecules.
>
>
> ### Response to question 2 in Questions:
>
> Thanks for the interesting question! Currently, there's a relatively small variance in molecule data lengths compared to text lengths. For instance, while text length varies from 30 to 500, most molecule sizes do not exceed 60. Therefore, the size of the molecules doesn't pose a significant challenge to computational efficiency. For extreme cases, the computation cost may slightly increase due to the increased input length.

---

> > ### Comment · Reviewer_3WHb · 2023-08-10
> > **Thanks for the rebuttal**
> >
> > The authors have addressed my questions. I'm maintaining my previous score and recommend acceptance.

---

> > > ### Author Response · Authors · 2023-08-11
> > >
> > > Thank you for your feedback! We remain available to address any further questions you may have.

---

### Official Review · Reviewer_M8Li · 2023-06-28

**Soundness:** 3 good
**Presentation:** 3 good
**Contribution:** 3 good
**Rating:** 6
**Confidence:** 4

**Summary:**

The paper proposes a new unified language model for graph and text data for instruction-based molecule zero0shot learning. The paper tries to address the problem of a supervised fine-tuning approach where labeled data by instruction tuning. The paper first treats both graph nodes and instruction tokens as input tokens. The paper then proposes a new relative position embedding to represent the edges in the graph and the relative position for text. The paper then retrains the model on Chembl and sometimes pm several tasks from MoleculeNet.The instructions are based on websites and databases. In a zero-shot setting, the model is compared against several baselines, including KVPLM, Galactica, and MoMu. The model is also compared to the graph-based models, including GCN, GAT, GIN, and Graphormer, in a supervised setting. The paper applies few-shot instruction-0based tuning for several datasets and compares it against KVPLM, MoMu, and GIN. The paper then conducts ablation and exploration studies.

**Strengths:**

1. The paper proposes a new unified text-graph T5-based model to unify molecular graphs and task instructions. The idea of relative position embedding for molecular graphs is interesting.
2. The paper achieves strong performance against several zero-shot baselines with fewer parameters. The paper also conducts experiments over multiple different tasks. The model also performs well on the few-shot settings. The paper shows its model design by conducting an ablation study to analyze the effectiveness of unified inputs and decoupling encoding. The paper also shows the effectiveness of pertaining over two different pertaining dataset. The paper also examines the robustness of instruction by GPT3.5 paraphrase. Finally, the paper ablates the explanation of instructions in downstream tasks.
3. The paper includes a very detailed appendix to explain some parts of the paper. The paper also visualizes attention on molecular to help readers understand the model.

**Weaknesses:**

1. The idea of encoding graphs with a seq2seq transformer has been introduced previously. This idea has been applied to graph-to-text tasks since 2021 (Ribeiro et al., 2021), despite using an unchanged T5 structure. One contribution of this paper is the new relative position embedding. However, the paper fails to include it in the ablation study.
2. The input of the model needs to be clarified. It would be better to include a sample input-output pair in the Appendix. Does the model use each node's chemical element as the molecular graph node's textual form? The datasets used for testing are hard to understand for readers who are not familiar with them. It would be better to include a small explanation for each testing dataset in the Appendix.
3. The paper fails to include a limitation section and an ethical section for potential social impact.

Ribeiro, L., Schmitt, M., Schutze, H., & Gurevych, I. (2021). Investigating Pretrained Language Models for Graph-to-Text Generation. In Proceedings of the 3rd Workshop on Natural Language Processing for Conversational AI (pp. 211–227). Association for Computational Linguistics.


**Questions:**

 Does the model use each node's chemical element as the molecular graph node's textual form?

**Limitations:**

The paper fails to include a limitation section. The paper needs to include a social impact section to discuss the potential misuse of the current system.

---

> ### Author Rebuttal · Authors · 2023-08-09
>
> ### Response to question 1 in weaknesses:
>
> **Graph-to-text tasks related work**
>
> Thanks for the reminding of related work. We will add this and other related work in the next version of the paper.
> The difference between the mentioned work and ours lies in the task objectives. The mentioned work processes serialized graph and translate it to the corresponding text. our task involves utilizing both molecular graph data and task-specific instructional text as input, with the objective of generating corresponding task outcomes as output.
>
> **Ablation study of the position embedding**
>
> We conducted an ablation study to assess the impact of our unified graph-text encoding method augmented with mixture position embedding. In this study (refer to Subsection 4.3 Ablation and Exploration Studies, Table 4), we introduced a simplified model labeled as "w.o. unifying." This model employs a GNN to derive graph features, which are then fed into the language model as tokens, without our mixture position embedding. The ablation experiment clearly demonstrates that this "w.o. unifying" model underperforms compared to GIMLET. This result underscores the significance of the unified graph-text encoding method that we have introduced.
>
> We further extend the ablation analysis on the position embedding by introducing the GIMLET-SMILES baseline. In GIMLET-SMILES, a regular T5 model is employed to process instructional text and sequential graph information, represented as SMILES sequences of molecules. To evaluate the performance of both GIMLET and GIMLET-SMILES, we employed instruction-based zero-shot testing on the Chembl zero-shot dataset. The experimental result is as follows:
>
> | Model         | Chembl zero-shot ROC-AUC $\uparrow$ |
> |---------------|------------------------------------|
> | GIMLET        | 0.7860                             |
> | GIMLET-SMILES | 0.7414                             |
>
> The results clearly indicate that the model utilizing the graph encoding method (GIMLET) surpasses the ablated GIMLET-SMILES. This ablation study further illustrates the advantages of our graph-encoding method.
>
>
> ### Response to question 2 in weaknesses (and question 1 in Questions):
>
> Sorry for the confusion, and your understanding is right. Each graph node corresponds to an atom within the molecule, identified by its respective chemical element. The representation of our input data is provided in Figure 1. We will include more detailed illustrations of data in the appendix in the next version of the paper.
>
> ### Response to question 3 in weaknesses:
>
> Thanks for your reminder. We intend to include a section on limitations and ethical considerations in the upcoming version of the paper.

---

> > ### Comment · Reviewer_M8Li · 2023-08-13
> > **Response to authors' rebuttal**
> >
> > Thanks for the rebuttal. The authors have answered most of my questions.  I'm maintaining my previous score.
> >
> > However, I want to clarify the position embedding ablation study part. So basically, I want to see a GIMLET with most of its original structures but exclude the position embedding part (eq3). It would be great if the authors could provide the results of it.

---

> > > ### Author Response · Authors · 2023-08-17
> > >
> > > Thank you for your feedback and the suggestion regarding the ablation study on position embeddings. In our previous ablated model, "w.o. unifying" and "GIMLET SMILES" were designed to exclude the unified position embedding method, while still preserving the capacity to enrich the model with molecular structure information by other methods. Following your recommendation, we explored another interesting baseline, "GIMLET w.o. P," which omits the position embeddings and disregards the structural information of the molecule. In this case, the model is only provided with the information of the set of molecule nodes. The result is as follows:
> > >
> > > | Model         | Chembl zero-shot ROC-AUC $\uparrow$ |
> > > |---------------|------------------------------------|
> > > | GIMLET        | 0.7860                             |
> > > | GIMLET w.o. P | 0.6227                             |
> > >
> > > It is not surprising that the performance drops significantly when the position embeddings for graph structure are removed. This occurs because the molecular structure information is disregarded, leaving the model with only the set of nodes. Deprived of sufficient information about the input molecule, the model struggles to perform the tasks effectively.

---

> > > > ### Comment · Reviewer_M8Li · 2023-08-17
> > > > **Response to authors' rebuttal**
> > > >
> > > > Thank you so much for your quick response! The authors have addressed my problems.

---

> > > > > ### Author Response · Authors · 2023-08-19
> > > > >
> > > > > Thank you for your feedback! We remain available to address any further questions you may have.

---

### Official Review · Reviewer_UK2h · 2023-07-04

**Soundness:** 2 fair
**Presentation:** 2 fair
**Contribution:** 2 fair
**Rating:** 4
**Confidence:** 4

**Summary:**

This paper proposes a unified language model for both graph and text data with two main tech contributions: (1) a unified graph-text transformer encoder with a distance-based joint position embedding to encode graphs, and (2) textual instructions to enhance transferability among tasks. Zero-shot tests on classification and regression tasks are conducted to show its effectiveness.

**Strengths:**

The high-level motivations of a unified graph-text model for molecular understanding and enhancing transferability with instructions are interesting.

**Weaknesses:**

1. It doesn’t make sense to evaluate the model on classification and regression tasks. The model structure is based on T5, which is for generation/translation tasks. In fact, the results in Table 1 show a huge performance gap between GIMLET and GNNs. Why not use those GNNs and Graphormers with better performance and fewer parameters? I think additional experiments of fine-tuning GIMLET on the train datasets of Bio-activity, Toxicity, and Pharmacokinetic tasks in Table 1 are needed to make a fair comparison with supervised GNNs.
2. Claims need to be further clarified. For example, in line 150-154, I don’t understand “the dense vectors encoded by GNN have a limited capacity to carry structure information”. According to the results in your Table 1, those dense vectors encoded by GNN actually carry more information than GIMLET. And “training the additional module is difficult due to the increased layers”, actually, in MoMu and CLAMP (cited in line 146), the graph and text encoders are two-tower structure like CLIP, which I don’t think increased layers.
3. The writing needs to be improved.


**Questions:**

Critical question: Do we really need a text-decoder model instead of conventional GNNs to perform classification and regression tasks?

**Limitations:**

See weakness part.

---

> ### Author Rebuttal · Authors · 2023-08-09
>
> ### Response to question 1 in Weaknesses and question 1 in Questions:
>
> **Why use the language model (T5)?**
>
> Thank you for your question regarding the task setting. This study focuses on instruction-based zero-shot learning for molecule property prediction. Our approach aims to predict properties by natural language task instructions, **without the need for supervision from training data**. This methodology aligns with the instruction-based zero-shot learning paradigm [1], as introduced in Section 2: Related Work - Instruction-based Zero-Shot Learning.
>
> In comparison to supervised training for molecule tasks, instruction-based zero-shot learning offers several key advantages. Firstly, it eliminates the need for task-specific labeled data, which can often be complex and expensive to annotate for molecules. Secondly, it allows the execution of new tasks without the resource and time-intensive process of retraining on new task samples. Lastly, it offers a more user-intuitive way of interacting with the model through natural language task descriptions, as opposed to relying solely on task samples. This is why instruction-based task execution has gained significant popularity not only in natural language processing [2] but also in the realm of visual-text multimodal tasks [3].
>
> To enable the model to perform novel molecule tasks **without the need for supervised training, task instructions for these new tasks are crucial**. By comprehending these instructions, the model can directly carry out the new tasks.
>
> The necessity to comprehend textual instructions underscores the imperative for utilizing a language model. In pursuit of this goal, we extend the capabilities of the existing T5 language model to encode both graph and text data.
>
> **Is the task format suitable for T5?**
>
> Regarding task type classification and regression, we unifiy the outputs of different tasks into the text modality. This is explained in Subsection 3.1: Problem Formulation and Framework Overview, and depicted in Figure 1. The outputs of diverse tasks are presented as text, such as "yes," "No," or "3.11." This approach unifies various tasks into conditioned text generation tasks, which aligns well with the nature of the T5-type model, an encoder-decoder model for conditional text generation.
>
> **Why not utilize GNNs?**
>
> The primary limitation is that graph-only models are unable to execute the instruction-based learning approach, involving both text and graph modalities. GIMLET, however, unifies the two modalities within a single model, enhancing its capacity for graph encoding and textual comprehension.
>
> **The performance gap between GIMLET and GNNs?**
>
> First, it is important to address that in Subsection 4.1 (Table 1, 2, and 3), the results of GIMLET and all the molecule-text model baselines are evaluated using instruction-based zero-shot learning. In contrast, the results of GNNs are obtained through supervised learning. We present the supervised learning results as a reference to gauge the difficulty of the tasks.
>
> To illustrate the model performance improvement when supervised training data is available, we conduct few-shot experiments in Subsection 4.2: Instructions-Based Few-Shot Fine-tuning. Our experiment demonstrates that with a small amount of training data, GIMLET's performance can be further enhanced.
>
> Additionally, we present the experimental results under the fully finetuned setting here:
>
> Supervised result (ROC-AUC $\uparrow$ ) over Bio-activity, Toxicity, and Pharmacokinetic tasks.
> |            | bace   | hiv    | muv    | Avg.bio | tox21  | toxcast | Avg.tox | bbbp   | cyp450 | Avg.pha |
> | ---------- | ------ | ------ | ------ | ------- | ------ | ------- | ------- | ------ | ------ | ------- |
> | GIN        | 0.701  | 0.753  | 0.718  | 0.724   | 0.740  | 0.634   | 0.687   | 0.658  | 0.821  | 0.739   |
> | Graphormer | 0.7760 | 0.7452 | 0.7061 | 0.7424  | 0.7589 | 0.6470  | 0.7029  | 0.7015 | 0.8436 | 0.7725  |
> | GIMLET     | 0.8280 | 0.7834 | 0.7267 | 0.7794  | 0.7676 | 0.6591  | 0.7134  | 0.7315 | 0.8809 |0.8062|
>
> Supervised performance (RMSE $\downarrow$) on Physicalchemical datasets.
>
> |            | ESOL  | lipo  | FreeSolv | Avg.phy |
> | ---------- | ----- | ----- | -------- | ------- |
> | GIN        | 1.243 | 0.781 | 2.871    | 1.632   |
> | Graphormer | 0.901 | 0.740 | 2.210    | 1.284   |
> | GIMLET     | 0.850 | 0.945 | 1.881    | 1.226   |
>
> It is evident that with full supervised training data, GIMLET's performance surpasses or is comparable to supervised baselines. This reveals the potential of GIMLET in supervised fine-tuning.
> ### Response to question 2 in weaknesses:
> Sorry for the confusion. In lines 150-154, we discuss a simple and common method for combining text with other modalities, i.e., the features obtained from the GNN are input into the language model as tokens. This simple method is also tested in our ablation study (Table 4) and is denoted as "w.o. unifying." The ablation experiment reveals that it performs worse than GIMLET.
>
> The purpose of lines 150-154 is to analyze the drawbacks of this simple method. First, the dense vectors encoded by the GNN have limited capacity to carry structural information as input for language models. Furthermore, the additional GNN is appended before the initial layer of the language model, resulting in increased layer count and potential issues.
>
> Our intention is not to refer to the GNN capacity in general case, or the contrastive baselines like MoMu. We will provide clearer explanations for the situation of these claims to avoid confusion in the future version.
>
> ### Response to question 3 in weaknesses:
>
> Thanks for pointing out! We will improve the writing in the future version.
>
> [1] Multitask Prompted Training Enables Zero-Shot Task Generalization. In ICLR 2022.
>
> [2] Finetuned Language Models are Zero-Shot Learners. In ICLR 2022.
>
> [3] Blip-2: Bootstrapping language-image pre-training with frozen image encoders and large language models. preprint 2023.

---

> > ### Comment · Reviewer_UK2h · 2023-08-17
> >
> > The authors have answered most of my questions. Thus, I would like to raise my score by one point.

---

> > > ### Author Response · Authors · 2023-08-19
> > >
> > > We sincerely appreciate your valuable feedback. We were wondering if there might be any additional concerns or unresolved issues that are preventing the paper from achieving a positive rating. We remain available to address any further questions you may have.
> > > Thank you for your time and consideration.

---

### Official Review · Reviewer_3HAC · 2023-07-06

**Soundness:** 3 good
**Presentation:** 3 good
**Contribution:** 3 good
**Rating:** 4
**Confidence:** 3

**Summary:**

This paper presents GIMLET, a unified graph-text model for instruction-based molecule zero-shot learning. The proposed model uses natural language instructions to tackle molecule-related tasks in a zero-shot setting. GIMLET overcomes existing limitations and significantly outperforms molecule-text models. The paper includes experimental results that demonstrate the superiority of GIMLET over molecule-text models. The paper also explores the use of pretraining tasks and the robustness of GIMLET to instructions. Overall, the paper provides a comprehensive approach to molecule zero-shot learning using natural language instructions.

**Strengths:**

This paper introduces GIMLET, a pioneering, integrated graph-text model specifically designed for instruction-based, zero-shot learning in molecular sciences. Leaping beyond the confines of current methodologies, GIMLET leverages natural language instructions to proficiently navigate molecule-related tasks in a zero-shot context. It surpasses the performance of conventional molecule-text models, reflecting substantial advancements in this field. Through rigorous experimental analysis, the superiority of GIMLET over conventional models is vividly illustrated.

**Weaknesses:**

1.One thing I'm not quite sure about is why the authors chose only a few molecule-text models as baselines, with MoMu and Galactica not even being specifically designed for predicting molecular properties. I understand that the authors might have done this because the model proposed in the paper is based on both molecules and text, but for a fair comparison, shouldn't other types of models (like those involving only molecules and not text) that predict molecular properties also be used as baselines? Otherwise, how can we highlight the advantage of incorporating text into the model for this task?

2.The paper uses a graph-text position encoding method to encode molecular graphs. Given that molecules already have their serialized representations, such as SMILES and SELFIES, isn't this approach superfluous and potentially introducing unnecessary computations and noise?

3.Based on the description in this paper, I can't clearly understand the specific definition of molecule zero-shot learning, which doesn't seem to differ much from other pre-training-fine-tuning training methods? Perhaps I misunderstood, could the authors provide a more precise explanation?

4.In Figure 1, the authors don't seem to have clearly described the specific meaning of different colored lines and boxes, which makes it difficult for readers to interpret this image.

**Questions:**

See weaknesses.

**Limitations:**

The authors have adequately addressed the limitations.

---

> ### Author Rebuttal · Authors · 2023-08-09
>
> Thanks for your questions. We would like to respond to question 3 first, which pertains to the fundamental aspect of the task our paper is working on.
>
> ### Response to question 3:
>
> Sorry for the confusion. In this work, we propose to investigate the feasibility of employing instructions to accomplish molecule tasks in a zero-shot setting. This approach aligns with the paradigm of instruction-based zero-shot learning [1], which we introduced in Section 2: Related Work - Instruction-based Zero-Shot Learning.
>
> **What is instruction?** The instructions, also referred to as prompts, refer to natural language descriptions of the tasks to be executed. For example, in the case of the text abstract generation task, the instruction could be "Write a summary for the following text." In our context, these are natural language instructions for molecule property prediction tasks. For instance, in the "toxicity to ARE pathway" task (from Tox21), the instruction is as follows: "Oxidative stress has been implicated in the pathogenesis of a variety of diseases ranging from cancer to neurodegeneration. The antioxidant response element (ARE) signaling pathway is important in the amelioration of oxidative stress. Is this molecule agonists of antioxidant response element (ARE) signaling pathway?"
>
> **What is instruction-based molecule zero-shot learning?** The instruction-based molecule zero-shot learning aims to enable model to perform unseen molecule tasks **without supervised training**. This is enabled by the provided task instructions for new tasks. By understanding the instructions, the model can perform the unseen tasks directly. For example, the "toxicity to ARE pathway" task mentioned above is not seen by the GIMLET before, but GIMLET is able to perform it without supervision by utilizing and understanding the task instruction. This approach eliminates the need for labeled data and leverages the textual knowledge available for downstream tasks.
>
> **How can we enable model to perform instruction-based molecule zero-shot learning?** It has been shown that language models exhibit strong instruction-based zero-shot performance after pretraining on a large scale of tasks with instructions [1]. For the molecule domain, we also do the instruction-based task pretraining for our graph-text model GIMLET. As illustrated in Figure 2 (left), we partition tasks into pretraining tasks and downstream testing tasks, without overlapping. After pretraining, we zero-shot-test model on the downstream tasks with task instructions provided in Subsection 4.1: Instruction-Based Zero-Shot Learning, which is our main experiment.
>
> **Difference between pretrain-finetuning method?** The main difference lies in the testing session, where instruction-based zero-shot learning does not require supervised training data, but executes the new tasks by only instructions.
>
>
> ### Response to question 1
>
> We employ molecule-text models as baselines due to our focus on the instruction-based zero-shot learning setting. In this context, the input comprises both molecule data and task instruction text, necessitating the utilization of molecule-text models rather than graph-only models.
>
> Besides the molecule-text models, we also have presented supervised results of graph-only models in Tables 1 and 3. These models include GCN, GAT, GIN, and Graphormer, serving as references for the supervised results.
>
> ### Response to question 2
>
>
>
>
> This is indeed an insightful question. While serialized representations of molecules, such as SMILES, can convey molecular information, they might fall short in accurately reflecting the actual structural features of the molecule's graph.
>
> To better illustrate the impact of employing graph representations on performance and speed, we conducted an additional experiment. We pretrained another T5 model using the same pretraining settings as GIMLET, but utilizing the SMILES representation of molecules rather than graph. This version is referred to as GIMLET-SMILES. We evaluate the performance of both GIMLET and GIMLET-SMILES using instruction-based zero-shot testing on Chembl zero-shot dataset. Additionally, we present the average FLOPs per data point as a metric to compare the computational efficiency. The experimental results are as follows:
>
> | Model | Chembl zero-shot ROC-AUC $\uparrow$  | Avg. FLOPs per data $\downarrow$  |
> |---- |  ----  | ----  |
> |GIMLET|	0.7860|	5.03E+09
> |GIMLET-SMILES|	0.7414|	5.12E+09
>
>
> It is evident that when utilizing the molecular graph, the model outperforms the use of molecule SMILES in instruction-based zero-shot learning tasks. Remarkably, the computational costs of the two models are nearly identical. This is due to GIMLET employing the same computational framework as the regular T5 model, with the graph-text mixture positional embeddings incurring costs equivalent to the normal T5 relative positional embeddings. There is no need for additional modules or computations within the model.
>
>
> ### Response to question 4:
>
> Sorry for the confusion. We suppose the color and lines you mentioned are the illustration of distance aware encoding, where lines with different colors refer to different distance values. In our upcoming versions, we intend to provide a more comprehensive explanation of the figure for clarity.
>
>
> [1] Sanh, V., Webson, A., Raffel, C., Bach, S. H., Sutawika, L., Alyafeai, Z., ... & Rush, A. M. (2022, April). Multitask Prompted Training Enables Zero-Shot Task Generalization. In ICLR 2022-Tenth International Conference on Learning Representations.

---

> > ### Author Response · Authors · 2023-08-19
> >
> > We sincerely hope that our response adequately addresses your questions. We remain available to address any further questions you may have.
> > Thank you for your time and consideration.

---

> ### Author Response · Authors · 2023-08-21
>
> Does our response adequately address your concern? If there are any remaining issues, we will continue to work on resolving them.

---

### Official Review · Reviewer_drQy · 2023-07-09

**Soundness:** 3 good
**Presentation:** 3 good
**Contribution:** 3 good
**Rating:** 5
**Confidence:** 3

**Summary:**

The paper addresses the challenge of molecule property prediction, especially the label insufficiency caused by costly lab experiments.
It uses  natural language instructions to handle molecule-related tasks in a zero-shot setting. The authors propose GIMLET, a model that unifies language processing for both graph and text data. GIMLET uses generalized position embedding to encode graph structures and instruction text without additional modules. It also decouples the encoding of the graph from task instructions, thereby improving graph feature generalization across different tasks. The authors create a dataset of over two thousand molecule tasks with corresponding instructions and pretrain GIMLET on these tasks, which allows the model to transfer effectively across tasks. GIMLET demonstrates good performance in instruction-based zero-shot learning, even closely matching supervised Graph Neural Network (GNN) models on tasks such as toxcast and muv.

**Strengths:**

- The paper is well-written and clearly presented;
- The authors create a comprehensive dataset for molecule tasks with instructions, which can be a valuable resource for future research.
- The ideas of leveraging instruction plus graph structures are novel ways for molecule property pretraining and the proposed unified GIMLET for graph and text data could contribute to the field of cheminformatics.
- The zero-shot performance of proposed methods is strong across baselines and tasks. Detailed ablations including pretraining, model design, and instructions are presented to illustrate the design choices.

**Weaknesses:**

- While GIMLET demonstrates strong performance in zero-shot learning, how could it perform with more supervised learning scenarios (beyond 256 or even with full training set) is questionable. The instruction-tuned LLM demonstrates strong generalization performance on single-task finetuning in NLP, is this also true for GIMLET remains a question;
- How could model size (only one size is presented), the scale of pretraining, and the diversity/quality of instruction / pretraining tasks affect GIMLET performance, it could also be valuable to see the performance of instruction-aided evaluation / qualitative examples on helf-on validation tasks (whether GIMLET could actually follow the instructions).

**Questions:**

- How model size, and the scale of pretraining (not only the domain) contribute to the final performance of GIMLET;
- Could GIMLET be extended to using frozen / parameter-efficient fine-tuned language models? / How to validate if the instruction-based pretraining / the proposed GIMLET model is scalable?

**Limitations:**

See above

---

> ### Author Rebuttal · Authors · 2023-08-09
>
> ### Response to question 1 in Questions and question 2 in Weaknesses:
>
> Thank you for your insightful question. We conduct experiments to illustrate the impact of varying pretraining scale and model size on GIMLET. To manipulate the pretraining scale, we explore different task numbers (tasks selected randomly) in the pretraining phase. Our evaluation includes instruction-based zero-shot performance on the large scale Chembl zero-shot testing set, as well as testing GIMLET across various model sizes on Bio-activity, Toxicity, and Pharmacokinetic tasks. The summarized results are presented below:
>
>  Zero-shot performance (ROC-AUC) over Chembl zero-shot
>
> | Pretraining task ratio | 1/4  | 1/2  | Full(1.1K)  |
> | ----   | ----  | ----  | ----  |
> GIMLET(T5-small) |	0.6650	 |	0.7343 |0.7860
> GIMLET(T5-base) |		0.6384 |		0.7318 |		0.8041
> GIMLET(T5-large) |		0.6921 |		0.7562	 |	0.8178
>
>  Zero-shot performance (ROC-AUC) over Bio-activity, Toxicity, and Pharmacokinetic tasks
> | -| bace   | hiv  | muv | Avg. bio | tox21  | toxcast| Avg.tox | bbbp| cyp450| Avg. pha |
> | ---- |  ----  | ----  | ----  | ----  | ----|----|----|----|----|----|
> |GIMLET(T5-small)	|0.6957	|0.6624	|0.6439|	0.6673	|0.6119|	0.5904|	0.6011|	0.5939|	0.7125|	0.6532
> GIMLET(T5-base)	|0.7240|	0.6636|	0.6322|	0.6733|	0.6136|	0.5811|	0.5974|	0.7087|	0.7174|	0.7131
> GIMLET(T5-large)|	0.6855|	0.6986|	0.6421|	0.6754|	0.5988|	0.5773|	0.5880|	0.6758|	0.7365|	0.7062
>
> From our experiment, we can draw the following conclusions:
>
> (a) The principal bottleneck in the current GIMLET framework appears to be the number of tasks, or instructions, used during pretraining. As the number of pretraining tasks increases, the performance of GIMLET consistently improves. Despite already incorporating 1.1k tasks for pretraining, there's still room for enhancement, especially considering the variability and complexity of molecule property prediction tasks. Thus, increasing the number of tasks is a primary direction for enhancing GIMLET's performance.
>
> (b) Enlarging the model size also leads to performance gains across most tasks. However, the improvements are relatively minor compared to increasing the scale of the dataset. This phenomenon can be attributed to the constrained size of pretraining instructions, which prevents full utilization of the capacity of larger models.
>
> In summary, both scaling up pretraining and increasing model size have the potential to enhance GIMLET's performance. However, in the current context, the primary bottleneck lies in the scale of pretraining tasks.
>
> ### Response to question 2 in Questions:
>
> This is indeed a key question. Our model architecture is adaptable for direct application to language models of varying scales, but the training method needs to be adjusted accordingly.
>
> First, the model architecture we have introduced, which includes mixture of distance-based position embeddings and casual attention between graphs and text, represents a general approach that empowers language models to effectively understand graph data. This approach is applicable to language models of various scales.
>
> Second, a significant aspect related to model scalability is the training method employed. While we trained the T5 model through continuous pretraining, for larger language models, opting for parameter-efficient fine-tuning techniques such as adapters would likely be a more optimal solution.
>
>
>
> ### Response to question 1 in Weaknesses:
>
> Thanks for the question. This study mainly focuses on zero-shot learning for molecule property prediction based on instructions, enabling generalization to unseen tasks without the requirement of labeled data, which typically necessitates expensive wet experiments. While we have shown that the performance of GIMLET can be enhanced through finetuning on few-shot examples in Subsection 4.2: Instructions-Based Few-Shot Finetuning, we also include the experimental results under the fully finetuned setting here:
>
>
> Supervised result (ROC-AUC $\uparrow$ ) over Bio-activity, Toxicity, and Pharmacokinetic tasks.
> |            | bace   | hiv    | muv    | Avg.bio | tox21  | toxcast | Avg.tox | bbbp   | cyp450 | Avg.pha |
> | ---------- | ------ | ------ | ------ | ------- | ------ | ------- | ------- | ------ | ------ | ------- |
> | GIN        | 0.701  | 0.753  | 0.718  | 0.724   | 0.740  | 0.634   | 0.687   | 0.658  | 0.821  | 0.739   |
> | Graphormer | 0.7760 | 0.7452 | 0.7061 | 0.7424  | 0.7589 | 0.6470  | 0.7029  | 0.7015 | 0.8436 | 0.7725  |
> | GIMLET     | 0.8280 | 0.7834 | 0.7267 | 0.7794  | 0.7676 | 0.6591  | 0.7134  | 0.7315 | 0.8809 |0.8062|
>
> Supervised performance (RMSE $\downarrow$) on Physicalchemical datasets.
>
> |            | ESOL  | lipo  | FreeSolv | Avg.phy |
> | ---------- | ----- | ----- | -------- | ------- |
> | GIN        | 1.243 | 0.781 | 2.871    | 1.632   |
> | Graphormer | 0.901 | 0.740 | 2.210    | 1.284   |
> | GIMLET     | 0.850 | 0.945 | 1.881    | 1.226   |
>
>
>
> It is evident that with a complete set of supervised training data, GIMLET's performance surpasses or is comparable to other supervised baselines. This reveals the potential of GIMLET in supervised fine-tuning.

---

> > ### Comment · Reviewer_drQy · 2023-08-18
> >
> > Thanks for the detailed rebuttals, which have answered many of my questions. And I raise my original rating.

---

> > > ### Author Response · Authors · 2023-08-19
> > >
> > > Thank you for your feedback on our paper. We kindly inquire whether there may exist any additional concerns or unresolved questions that might be impeding the paper's attainment of a higher rating. We remain available to address any further questions you may have.

---

### Decision · Program_Chairs · 2023-09-21

**Decision:**

Accept (poster)

**Comment:**

This paper presents GIMLET, a unified graph-text model for instruction-based molecule zero-shot learning. The proposed model uses natural language instructions to tackle molecule-related tasks in a zero-shot setting. GIMLET overcomes existing limitations and significantly outperforms molecule-text models. The paper includes experimental results that demonstrate the superiority of GIMLET over molecule-text models. The paper also explores the use of pretraining tasks and the robustness of GIMLET to instructions.

Overall, the paper provides a comprehensive approach to molecule zero-shot learning using natural language instructions.